# Lipocalin-2 is an anorexigenic signal in primates

**Peristera-Ioanna Petropoulou[1], Ioanna Mosialou[1], Steven Shikhel[1], Lihong Hao[2], Konstantinos Panitsas[1†], Brygida Bisikirska[1], Na Luo[1], Fabiana Bahna[3], Jongho Kim[4], Patrick Carberry[4], Francesca Zanderigo[5,6], Norman Simpson[5], Mihran Bakalian[5], Suham Kassir[5,6], Lawrence Shapiro[3], Mark D Underwood[5,6], Christina M May[7], Kiran Kumar Soligapuram Sai[8], Matthew J Jorgensen[7], Cyrille B Confavreux[9], Sue Shapses[2,10], Blandine Laferrère[11,12], Akiva Mintz[4], J John Mann[4,5,6], Mishaela Rubin[11], Stavroula Kousteni[1]\***

[1]Department of Physiology and Cellular Biophysics, Columbia University Medical Center, New York, United States; [2]Department of Nutritional Sciences, Rutgers University, New Brunswick, United States; [3]Department of Biochemistry and Molecular Biophysics, Columbia University, New York, United States; [4]Department of Radiology, Columbia University Medical Center, New York, United States; [5]Department of Psychiatry, Columbia University Medical Center, New York, United States; [6]Molecular Imaging and Neuropathology Area, New York State Psychiatric Institute, New York, United States; [7]Department of Pathology, Section on Comparative Medicine, Wake Forest School of Medicine, Winston-Salem, United States; [8]Department of Radiology, Wake Forest School of Medicine, Winston-Salem, United States; [9]INSERM UMR1033-Université de Lyon-Hospices Civils de Lyon, Lyon, France; [10]Department of Medicine, Rutgers - RWJ Medical School, Rutgers University, New Brunswick, United States; [11]New York Obesity Nutrition Research Center, Columbia University, New York, United States; [12]Department of Medicine, Division of Endocrinology, Columbia University Irving Medical Center, New York, United States

**\*For correspondence:**
sk2836@columbia.edu

**Present address:** [†]1889 Jefferson Center for Population Health, Jefferson College of Population Health, Thomas Jefferson University, Johnstown, United States

**Competing interests:** The authors declare that no competing interests exist.

**Abstract** In the mouse, the osteoblast-derived hormone Lipocalin-2 (LCN2) suppresses food intake and acts as a satiety signal. We show here that meal challenges increase serum LCN2 levels in persons with normal or overweight, but not in individuals with obesity. Postprandial LCN2 serum levels correlate inversely with hunger sensation in challenged subjects. We further show through brain PET scans of monkeys injected with radiolabeled recombinant human LCN2 (rh-LCN2) and autoradiography in baboon, macaque, and human brain sections, that LCN2 crosses the blood-brain barrier and localizes to the hypothalamus in primates. In addition, daily treatment of lean monkeys with rh-LCN2 decreases food intake by 21%, without overt side effects. These studies demonstrate the biology of LCN2 as a satiety factor and indicator and anorexigenic signal in primates. Failure to stimulate postprandial LCN2 in individuals with obesity may contribute to metabolic dysregulation, suggesting that LCN2 may be a novel target for obesity treatment.

## Introduction

Obesity is a global epidemic that results in millions of deaths every year; a chronic disease associated with other serious and chronic conditions including type 2 diabetes, coronary artery disease, stroke, cancer, and depression amongst others (*Heymsfield and Wadden, 2017*). Obesity affects adults and children and is linked to seven of the top ten leading causes of death and disability in the

**eLife digest** Obesity has reached epidemic proportions worldwide and affects more than 40% of adults in the United States. People with obesity have a greater likelihood of developing type 2 diabetes, cardiovascular disease or chronic kidney disease. Changes in diet and exercise can be difficult to follow and result in minimal weight loss that is rarely sustained overtime. In fact, in people with obesity, weight loss can lower the metabolism leading to increased weight gain. New drugs may help some individuals achieve 5 to 10% weight loss but have side effects that prevent long-term use.

Previous studies in mice show that a hormone called Lipocalin-2 (LCN2) suppresses appetite. It also reduces body weight and improves sugar metabolism in the animals. But whether this hormone has the same effects in humans or other primates is unclear. If it does, LCN2 might be a potential obesity treatment.

Now, Petropoulou et al. show that LCN2 suppressed appetite in humans and monkeys. In human studies, LCN2 levels increased after a meal in individuals with normal weight or overweight, but not in individuals with obesity. Higher levels of LCN2 in a person's blood were also associated with a feeling of reduced hunger. Using brain scans, Petropoulou et al. showed that LCN2 crossed the blood-brain barrier in monkeys and bound to the hypothalamus, the brain center regulating appetite and energy balance. LCN2 also bound to human and monkey hypothalamus tissue in laboratory experiments. When injected into monkeys, the hormone suppressed food intake and lowered body weight without toxic effects in short-term studies.

The experiments lay the initial groundwork for testing whether LCN2 might be a useful treatment for obesity. More studies in animals will help scientists understand how LCN2 works, which patients might benefit, how it would be given to patients and for how long. Clinical trials would also be needed to verify whether it is an effective and safe treatment for obesity.

U.S. (*National Center for Health Statistics (US), 2016*). There are limited effective medical treatment options for long-term weight loss mainly due to our limited understanding of energy homeostasis—the mechanism that sustains weight by matching energy intake to energy expenditure over time (*Schwartz et al., 2017*). In individuals with longstanding obesity, the body responds to long-term weight loss by a reduction in metabolic rate, favoring weight regain (*Fothergill et al., 2016*; *Rosenbaum et al., 2010*). Diet and exercise programs have high relapse rates and available pharmacotherapies have limited effectiveness, with safety concerns and poor tolerability (American College of Cardiology/American *American College of Cardiology/American Heart Association Task Force on Practice Guidelines, Obesity Expert Panel, 2013, 2014*; *Daubresse and Alexander, 2015*).

Lipocalin-2 (LCN2) is an endogenous hormone found in mice and humans (*Liu et al., 2018*; *Rucci et al., 2015*), secreted by osteoblasts and which suppresses food intake in mice (*Mosialou et al., 2017*). Long-term LCN2 administration to lean and obese mice suppresses appetite and body weight gain without loss of effect over time, and improves whole body glucose metabolism while at the same time increasing energy expenditure. Therefore, LCN2 overcomes the inherent compensatory decrease in energy expenditure that develops following a sustained decrease in food intake (*Mosialou et al., 2017*). Moreover, LCN2 acts as a satiety signal that is upregulated after feeding in mice to limit food intake. Its anorexigenic mechanism of action relies on its ability to cross the blood-brain barrier (BBB) and activate the melanocortin four receptor (MC4R)-dependent pathway, one of the most potent currently known regulators of obesity (*Mosialou et al., 2017*). Heterozygous mutations in MC4R are the commonest cause of monogenic obesity, affecting approximately 0.1% of the population (*Farooqi et al., 2003*).

Based on genetic, molecular, and biochemical studies in mice (*Mosialou et al., 2017*; *Rached et al., 2010*) we sought to determine whether the postprandial regulation and hypothalamic action of LCN2 is conserved in humans and non-human primates and whether the systemic administration of LCN2 in primates induces appetite suppression.

## Results

### Serum LCN2 levels are postprandially increased in individuals with normal and overweight but not in individuals with obesity or with severe obesity

To assess the postprandial regulation of serum LCN2, we used data from four separate studies where healthy individuals with normal weight, overweight, obesity, and severe obesity were challenged with a meal after an overnight fast. In Study 1 with young healthy lean women (BMI: $21.8 \pm 0.6$ Kg/m$^2$; *Supplementary file 1A*), analysis revealed a tendency for increase of circulating LCN2 levels with time ($F_{7, 70}=3.07$, p=0.065; *Figure 1A*), although not significant. LCN2 serum concentration at baseline (t = 0 min) trended to differ from the one at t = 30 min ($F_{1, 10}=3.8$, p=0.080), t = 45 min ($F_{1, 10}=4.6$, p=0.058), t = 60 min ($F_{1, 10}=4.87$, p=0.052), and t = 90 min ($F_{1, 10}=3.9$, p=0.076), a similar magnitude of postprandial upregulation to what was previously reported (*Paton et al., 2013*). Interestingly, postprandial LCN2 serum levels (mean concentration at each timepoint) were robustly inversely correlated with hunger scores (mean hunger scores at each timepoint) of the challenged subjects (Spearman r = −0.98, p=0.0004) after the consumption of the liquid meal (*Figure 1B*). Serum LCN2 levels peaked at 45 min after meal ingestion, increasing by 16% (*Supplementary file 1A*, *Figure 1—figure supplement 1, A*).

Similarly, postprandial circulating LCN2 levels were significantly altered with time ($F_{2, 16}=27.87$, p=0.002) in a separate, second study of young healthy lean women (BMI: $20.8 \pm 0.5$ Kg/m$^2$; *Supplementary file 1A*). Specifically, serum LCN2 concentration at t = 60 min ($F_{1, 8}=59.64$, p=0.002) and t = 105 min ($F_{1, 8}=15.36$, p=0.009; *Figure 1C*) were significantly increased from baseline. Here serum LCN2 levels peaked at 60 min increasing by 54% (*Supplementary file 1A*, *Figure 1—figure supplement 1, G*).

The third study included 47 subjects, 28 women and 19 men, with overweight and/or obesity (BMI: $28.7 \pm 0.5$ Kg/m$^2$). The whole cohort consisted of 30 subjects (18 women and 12 men) with overweight (BMI = $26.4 \pm 0.3$ Kg/m$^2$) and 17 subjects (10 women and 7 men) with obesity (BMI = $32.7 \pm 0.4$ Kg/m$^2$). Contrary to lean groups, LCN2 significantly decreased after the meal challenge ($F_{5, 215}=2.61$, p=0.026; *Figure 1D*).

Interestingly, based on their postprandial LCN2 response, this initial cohort could be divided into two subgroups (*Figure 1E*). The first group (n = 25) included responders (R), that is, subjects that had a 'positive' postprandial LCN2 response with elevated LCN2 levels in multiple timepoints after the meal. The second group (n = 22) consisted of non-responders (NR), that is, subjects that had a 'negative' postprandial LCN2 response with decreased LCN2 levels after the meal challenge.

Responders showed a trend, though not statistically significant, toward a 12% increase in serum LCN2 levels 60 min after meal consumption ($F_{1, 21}=3, 24$, p=0.086; *Figure 1E*; *Supplementary file 1B*; *Figure 1—figure supplement 2A*). The inverse correlation between postprandial LCN2 serum levels and hunger scores was attenuated compared to that of Study 1 (Spearman r = −0.66, p=0.33; *Figure 1F*). On the other hand, non-responders showed decreased LCN2 levels at all timepoints examined, reaching a nadir 60 min after the meal, with a 19% reduction ($F_{1,21}=37.08$, p<0.0001; *Supplementary file 1B*). Non-responders trended to have a significantly larger waist circumference (*Supplementary file 1B*). Higher values for BMI, body fat, diastolic blood pressure, and fasting serum glucose and LCN2 levels were also observed in the non-responders but did not reach statistical significance (*Supplementary file 1B*).

When the 47-subject mixed cohort was analyzed by sex, subjects could again be divided into responders and non-responders, based on their postprandial LCN2 response. Women (*Figure 1G*; *Figure 1—figure supplement 2, B*) and men (*Figure 1J*; *Figure 1—figure supplement 2, C*) responders showed a trend, though not statistically significant, toward a 10% and 15% increase in serum LCN2 levels 60 min after meal consumption, respectively (*Supplementary file 1B*). The inverse correlation between postprandial LCN2 serum levels and hunger sensation found in women with normal weight (in Study 1), was not present in these overweight/obese groups of women (*Figure 1H*; *Figure 1—figure supplement 2, E*) or men (*Figure 1K*; *Figure 1—figure supplement 2, F*). On the other hand, non-responders showed a significant LCN2 decrease postprandially at all timepoints examined, reaching a nadir at 60 min for women ($F_{1,12}=36.9$, p<0.0001; *Figure 1G*) and at 90 min for men ($F_{1,8}=25.21$, p=0.001; *Figure 1J*).

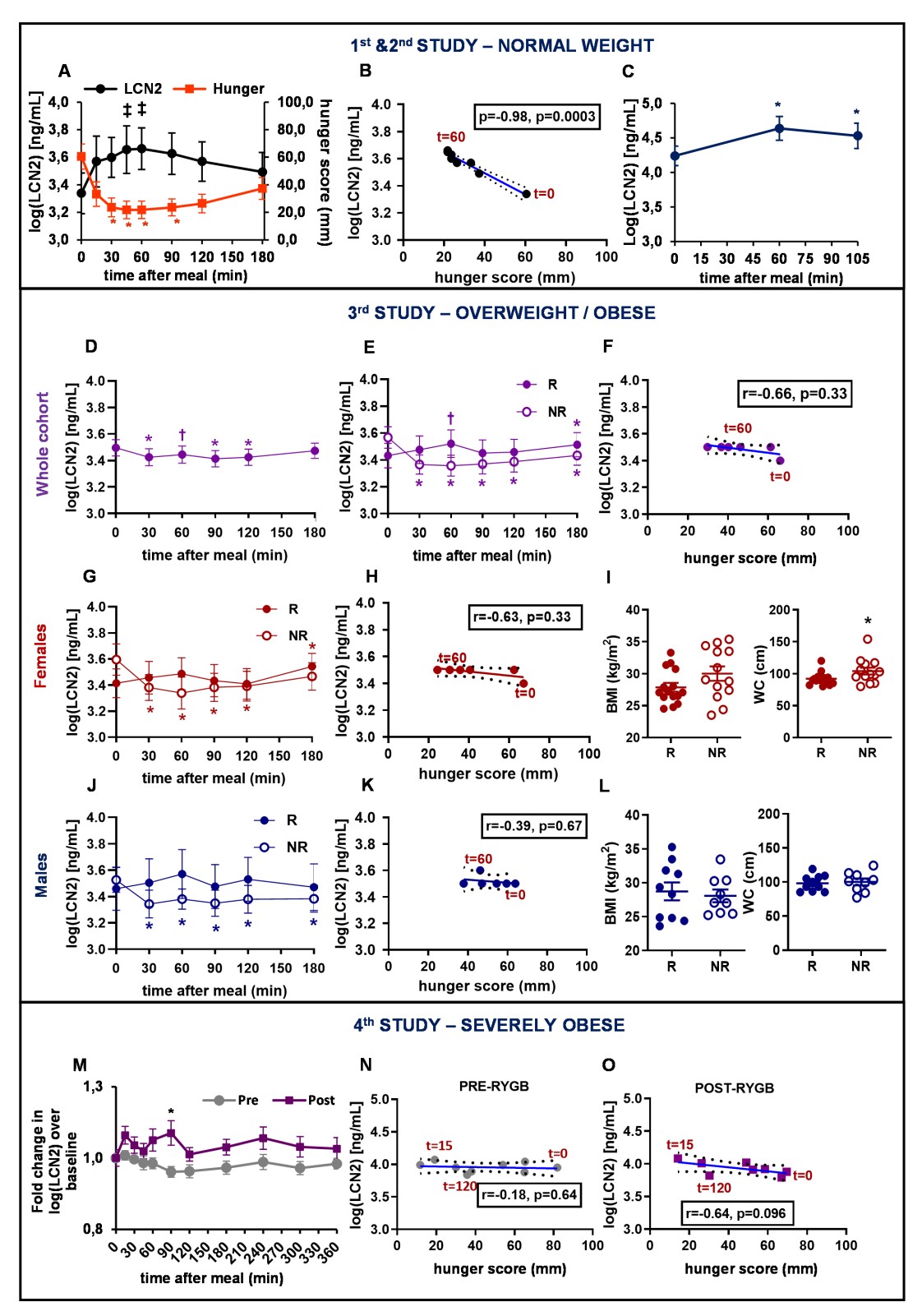

**Figure 1.** Serum LCN2 levels are postprandially increased in individuals with normal weight and overweight but not in individuals with obesity or with severe obesity. (A–B) Study 1: (A) Serum LCN2 levels and hunger and (B) Spearman correlation between serum LCN2 levels and hunger in normal-weight women (n = 11). (C) Study 2: serum LCN2 levels in normal-weight women (n = 9). (D–L) Study 3: (D–E) serum LCN2 levels in (D) all subjects (n = 47) and (E) subcategories of the cohort classified according to their postprandial response in raising LCN2 (R [n = 25]; NR [n = 22]). (F) Spearman

*Figure 1 continued on next page*

*Figure 1 continued*

correlation between serum LCN2 levels and hunger in the responders of the mixed cohort. (G) Serum LCN2 levels, (H) Spearman correlation between serum LCN2 levels and hunger and (I) BMI and waist circumference in female R (n = 15) and NR (n = 13) individuals. (J) Serum LCN2 levels, (K) Spearman correlation between serum LCN2 levels and hunger and (L) BMI and waist circumference in male R (n = 10) and NR (n = 9) individuals. (M–O) Study 4: (M) Fold change in serum LCN2 levels in female and male individuals with obesity, before (pre) and after (post) gastric bypass (n = 12). The asterisk denotes the difference before and after surgery at the indicated timepoint. (N–O) Spearman correlation between serum LCN2 levels and hunger in individuals with severe obesity (N) before and (O) after bariatric surgery. Values represent mean ± SEM. * indicates p<0.05, ‡ indicates p<0.06, and † indicates p<0.1 of each timepoint versus baseline, unless otherwise stated. 'Serum LCN2' represents log-transformed postprandial levels and 'hunger' represents hunger scores BMI = basic metabolic Index, LCN2 = Lipocalin-2, R = responders (elevated LCN2 levels in multiple timepoints after the meal), NR = non-responders (reduced LCN2 levels after the meal), RYGB = Roux en-Y Gastric Bypass.

The online version of this article includes the following source data and figure supplement(s) for figure 1:

**Source data 1.** Serum LCN2 levels are postprandially increased in individuals with normal weight and overweight but not in individuals with obesity or with severe obesity.

**Figure supplement 1.** Postprandial LCN2, but not GLP-1 or insulin, is correlated with hunger in women with normal weight.

**Figure supplement 1—source data 1.** Postprandial LCN2, but not GLP-1 or insulin, is correlated with hunger in women with normal weight.

**Figure supplement 2.** Postprandial LCN2, but not GLP-1 or insulin, is showing a decreased response in people with overweight.

**Figure supplement 2—source data 1.** Postprandial LCN2, but not GLP-1 or insulin, is showing a decreased response in people with overweight.

**Figure supplement 3.** Correlation of postprandial LCN2 with hunger is ameliorated in individuals with normalized weight, only after bariatric surgery.

**Figure supplement 3—source data 1.** Correlation of postprandial LCN2 with hunger is ameliorated in individuals with normalized weight, only after bariatric surgery.

Whereas women NRs had significantly higher waist circumference (*Figure 1I*) and showed a trend toward higher BMI, body fat, serum glucose, diastolic blood pressure (*Supplementary file 1B*), men NRs did not show any major differences in BMI, waist circumference (*Figure 1L*) or any other parameters (*Supplementary file 1B*).

The Study 4 included individuals with severe obesity, studied before and after Roux-en-Y gastric bypass surgery. The initial BMI of $47.4 \pm 1.9$ kg/m$^2$ was reduced to $29.6 \pm 1.8$ kg/m$^2$ one year after the surgery (*Stano et al., 2017*). Baseline fasting LCN2 levels were marginally decreased after surgery (*Supplementary file 1C*, *Figure 1—figure supplement 3A,C*). Postprandial levels of serum LCN2 were rather decreased before surgery ($F_{10, 109}=1.4$, p=0.253) and trended to be significantly increased after surgery ($F_{10, 107}=1.97$, p=0.079), suggesting re-sensitization of these subjects after normalization of BMI. Similar to the overweight and obese non-responders of the previous study, pre-surgery postprandial circulating levels of LCN2 showed a 19% decrease from baseline at 90 min ($F_{1,11}=6.54$, p=0.026) after the ingestion of the liquid meal (*Supplementary file 1C*). Interestingly, post-surgery, postprandial concentrations of LCN2 changed to the opposite direction showing a 42% increase at 15 min ($F_{1, 10}=7.54$, p=0.023) and a trend, though not statistically significant, toward 59% increase from baseline at 90 min ($F_{1, 11}=4.32$, p=0.065) after ingestion of the meal (*Supplementary file 1C*). Furthermore, Roux-en-Y gastric bypass significantly affected ($F_{1, 220}=5.89$, p=0.024) the observed difference in LCN2 levels at 90 min before and after surgery (*Figure 1M*). Of note, while postprandial LCN2 concentration did not correlate with hunger score before surgery (Spearman r = −0.18, p=0.64; *Figure 1N*; *Figure 1—figure supplement 3B*), there was an association, albeit non-significant after surgery (Spearman r = −0.64, p=0.096; *Figure 1O*; *Figure 1—figure supplement 3D*).

In order to place in context the regulation of postprandial LCN2 serum levels and its association with hunger, to those of other feeding-regulating hormones, we measured glucagon-like peptide 1 (GLP-1) and insulin circulating concentrations. In the normal-weighted cohorts (1st and 2nd Study) LCN2 showed a postprandial response similar in magnitude to that of GLP-1 (*Figure 2A–D* and *Figure 1—figure supplement 1A–B, G–H*). In both studies circulating insulin showed higher postprandial upregulation than LCN2 (*Figure 2A,C* and *Figure 1—figure supplement 1C,I*). However, the total response of LCN2 was significantly lower than GLP-1 (p=0.035; *Figure 2B* and *Figure 1—figure supplement 1A–B*) in Study 1, but not in study 2 (p=0.385; *Figure 2D* and *Figure 1—figure supplement 1G–H*). In study 1, LCN2 was the postprandial protein with the highest inverse correlation with hunger score (*Figure 1—figure supplement 1D–F*); GLP-1 was also inversely correlated with hunger, yet less strongly (*Figure 1—figure supplement 1E*). We did not find any correlation

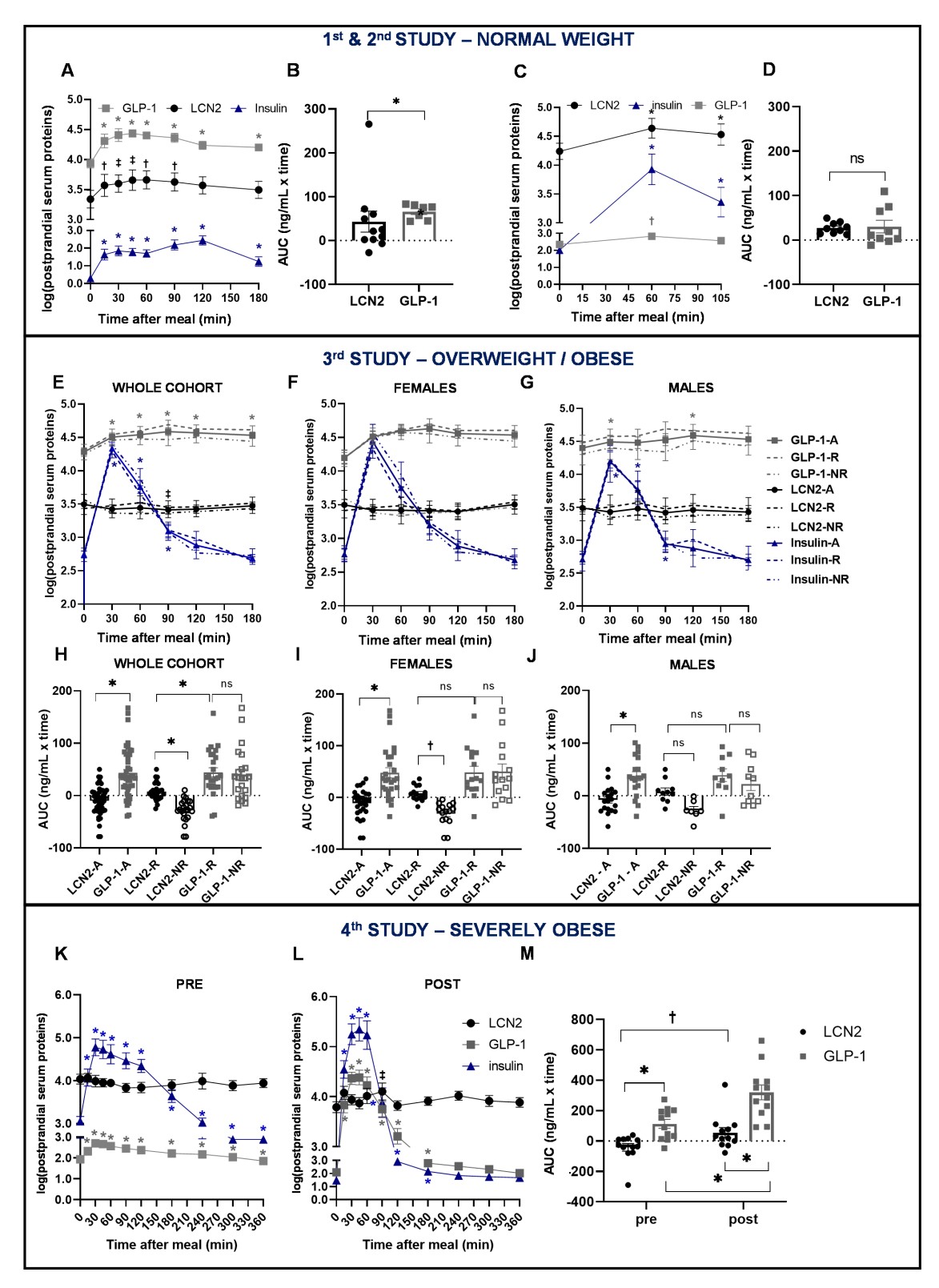

**Figure 2.** Similar postprandial regulation of serum LCN2 and GLP-1 levels in subjects with normal weight but not in subjects with overweight or obesity. (A–B) Study 1: (A) serum LCN2, GLP-1, and insulin levels of n = 11 normal-weight women and (B) area under the curve comparison for LCN2 and GLP-1. (C–D) Study 2: (C) serum LCN2, GLP-1, and insulin levels of n = 9 normal-weight women and (D) area under the curve comparison for LCN2 and GLP-1. (E–J) Study 3: (E) serum LCN2, GLP-1, and insulin levels of n = 47 overweight and obese subjects (whole, sex-mixed cohort), (F) of n = 28 overweight

*Figure 2 continued on next page*

*Figure 2 continued*

and obese women and (G) of n = 19 overweight and obese men and subcategorization of the cohort to responders and non-responders. Continuous lines were used for the whole, sex-mixed cohort (-A), the dashed line for the responders (-R) and the dash-and-dots line for the non-responders (-NR). Symbols mark the significant differences between each timepoint and baseline. (H) Area under the curve comparison for LCN2 and GLP-1 of the sex-mixed cohort, (I) women and (J) men. (K–M) Study 4: serum LCN2, GLP-1, and insulin levels of n = 12 obese subjects before/pre and (L) after/post gastric bypass. (M) Area under the curve comparison for LCN2 and GLP-1 pre- and post-gastric bypass surgery. Values represent mean ± SEM. * indicates p<0.05, ‡ indicates p<0.06 and † indicates p<0.1 of each timepoint versus baseline. 'Serum LCN2, GLP-1 and insulin' represent log-transformed postprandial levels. The units for log LCN2 and GLP-1 concentrations are in ng/mL, whereas for insulin in mIU/mL. LCN2 = Lipocalin-2, GLP-1 = Glucagon like peptide 1, RYGB = Roux en-Y Gastric Bypass.

The online version of this article includes the following source data for figure 2:

**Source data 1.** Similar postprandial regulation of serum LCN2 and GLP-1 levels in subjects with normal weight but not in subjects with overweight or obesity.

between insulin levels and hunger in this cohort (*Figure 1—figure supplement 1F*). For this reason, we more closely compared total responses of LCN2 and GLP-1.

In study 3, the total GLP-1 response was significantly higher than LCN2 (p<0.0001). For consistency purposes, we also analyzed GLP-1 and serum levels in responders and non-responders, although segregation in these two groups was based on LCN2 serum levels. The response of LCN2 was significantly different between responders and non-responders (p=0.014; *Figure 2H*) and this was more pronounced in females (*Figure 2I*), than males (*Figure 2J*). GLP-1 or insulin response between responders and non-responders was not in the opposite direction, as in the case of LCN2 (*Figure 2E-G*; *Figure 1—figure supplement 2, G–I,M–O*). Within the responders, an inverse correlation with hunger was present for insulin (*Figure 1—figure supplement 2, P–R*) but not for LCN2 (*Figure 1—figure supplement 2, D–F*) or GLP-1 (*Figure 1—figure supplement 2, J–L,*).

In study 4, the improvement of body weight and BMI after gastric bypass surgery was accompanied by a large increase in postprandial GLP-1 and insulin concentrations and to a lesser extent in LCN2 concentration (*Figure 2K-L*). GLP-1 response was higher than that of LCN2, both before (pre) and after (post) bariatric surgery (*Figure 2K-L*). While the GLP-1 response was significantly increased after the surgery, LCN2 only showed a tendency for increase (*Figure 2M*). GLP-1 and insulin showed a strong inverse correlation with hunger both before (*Figure 1—figure supplement 3, F,J*, respectively) and after the bariatric surgery (*Figure 1—figure supplement 3, H,L*, respectively). In contrast to LCN2, which did not correlate with hunger scores before surgery (*Figure 1—figure supplement 3, B*), tended to inversely correlate with it after the surgery (*Figure 1—figure supplement 3, D*), although not significantly.

Combined, our studies in humans show a postprandial increase in circulating LCN2 levels in humans with normal weight, which notably correlates with a drop in hunger sensation in the same individuals. Furthermore, subjects with overweight or obesity lose postprandial regulation of LCN2 and this may be a new mechanism of resistance that contributes to obesity.

## LCN2 crosses the blood-brain barrier of vervets and binds to the hypothalamus of human, baboon, and rhesus macaque brain sections

Next, we examined whether the mechanism of action of LCN2 is conserved in primates. As a first approach we evaluated whether [$^{124}$I] rh-LCN2 crosses the blood-brain barrier in non-human primates. Combined analysis of MRI and PET representative images of vervet monkey brain demonstrated an initial peak of activity throughout the brain during the first 30 s after the end of intravenous administration of [$^{124}$I] rh-LCN2 that is characteristic of BBB permeability. The sagittal, coronal, and axial MRI T1-weighted template images show a volume of interest (VOI) in the anatomical area of the hypothalamus where there is an indication of tracer binding although it may partially be spillover from an adjacent area outside the brain that also shows substantial tracer uptake (*Figure 3A-I*, *Figure 3—figure supplement 1A–M*).

PET acquisition was repeated using a chase/blocking paradigm to determine whether there is specific binding of [$^{124}$I] rh-LCN2 in the hypothalamus. The results from the chase experiment—standard uptake values at every timepoint—were compared to those of the no-chase experiment, by using the same procedure, software and atlas. Infusion of the MC4R receptor ligand, α-MSH, 15 min after [$^{124}$I] rh-LCN2 did not seem to affect the tracer uptake in the thalamic region (*Figure 3J and L*),

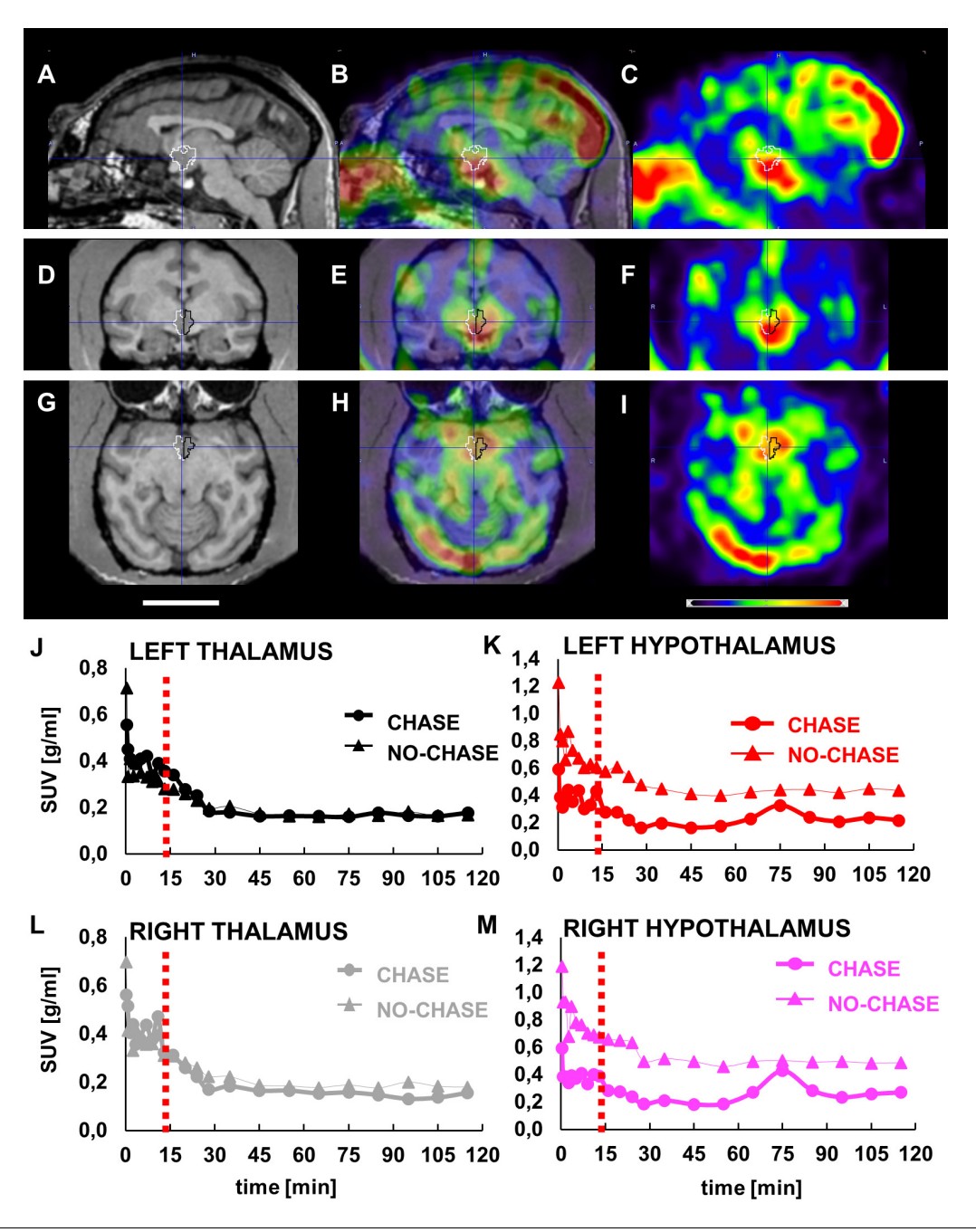

**Figure 3.** LCN2 crosses the blood-brain barrier of vervets. (A, D, G) MRI, (B, E, H) PET/MRI and (C, F, I) PET representative images of monkey brain 30 s after infusion of [$^{124}$I]-rh-LCN2. (A, B, C) Sagittal, (D, E, F) coronal, and (G, H, I) axial MRI T1-weighted template images (Invia19) demonstrate the volume of interest (VOI) and the anatomy of the hypothalamus (outlined with white and black line). (J–M) Time-activity curves (TACs) for the (J) left thalamus and (K) hypothalamus and (L) right thalamus, and (M) hypothalamus in a chase and a no-chase experiment in the same animal; TACs are reported in standard uptake value (SUV) units.

The online version of this article includes the following source data and figure supplement(s) for figure 3:

**Source data 1.** LCN2 crosses the blood-brain barrier of vervet monkeys.

**Figure supplement 1.** LCN2 crosses the blood-brain barrier of vervet monkeys.

**Figure supplement 1—source data 1.** LCN2 crosses the blood-brain barrier of vervet monkeys.

whereas it did reduce uptake compared with the no-chase condition in the hypothalamus (*Figure 3K and M*). We observed a 6.3% and 5.7% difference in the standard uptake value (SUV) in the left and right thalamus respectively, and a 49.8% and 51.2% reduction in the left and right hypothalamus. These results indicate that [$^{124}$I] rh-LCN2 penetrates the BBB and shows specific binding defined by displacement with α-MSH in the hypothalamus but not in the thalamus.

To further prove that LCN2 can bind to the hypothalamic feeding center of primates and to also exclude the possibility, inherent to PET studies, that a spillover signal from outside the brain may confound the findings, we examined LCN2 binding to brain sections where no such potential confounder exists. Rhesus macaque, baboon, and human brain sections containing the hypothalamus were incubated with either [$^{125}$I] rh-LCN2 alone or in the presence of excess unlabeled LCN2 or α-MSH to assess the specificity of binding. [$^{125}$I] rh-LCN2 binding was observed in the hypothalamic area of the baboon (*Figure 4A*; *Figure 4—figure supplement 1B,F*) and the rhesus macaque (*Figure 4—figure supplement 1, C–D*).

In the baboon, the specificity of binding was confirmed by the use of unlabeled LCN2 which blocked part of the [$^{125}$I] rh-LCN2 binding (*Figure 4B*). Specific binding was observed in the paraventricular nucleus of the hypothalamus (PVN) and both the dorsomedial (DM) and ventrolateral (VL) nuclei of the human hypothalamus, all areas where MC4R is expressed (*Figure 4C*, and *Figure 4—*

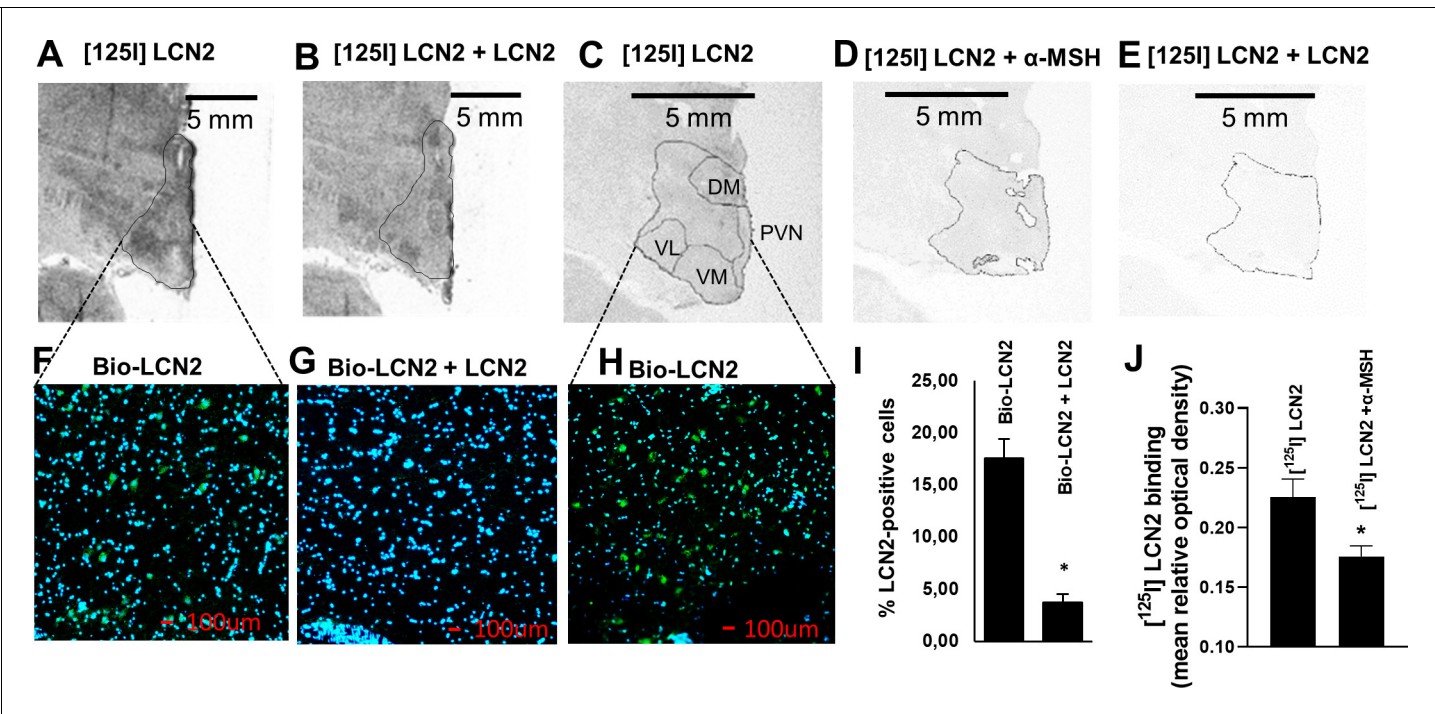

**Figure 4.** LCN2 binds to the hypothalamus of human, baboon, and rhesus macaque brain sections. (A–B) Autoradiographic images showing (A) [$^{125}$I] rh-LCN2 binding and (B) blocking of [$^{125}$I] rh-LCN2 binding with not radiolabeled rh-LCN2 on the baboon hypothalamus; the hypothalamic area is outlined with a black line. (C–E) Autoradiographic images showing (C) [$^{125}$I] rh-LCN2 binding, (D) blocking of [$^{125}$I] rh-LCN2 binding with α-MSH, and (E) blocking of [$^{125}$I] rh-LCN2 binding with not radiolabeled rh-LCN2 on the human hypothalamus. (F–G) Binding of biotinylated LCN2 to the hypothalamic area (outlined in A, B from baboon brain sections) in the (F) absence or (G) presence of hundred-fold excess of non-biotinylated LCN2 and (I) quantitation of LCN2-positive cells in both conditions (as percent of total cells in each field of view; n = 1 brain section and n = 4 fields of view for (F) and n = 2 for (G)). Bar graphs were obtained from a single brain section and therefore depict qualitative representations of binding. (H) Binding of biotinylated LCN2 to the hypothalamic area (outlined in C-E) from the human brain. (J) Quantification of specific [$^{125}$I] rh-LCN2 binding to human brain sections (n = 3). Values are mean ± standard deviation of the mean. DM = dorsomedial, PVN = paraventricular nucleus of the hypothalamus, VM = ventromedial, VL = ventrolateral nucleus of the hypothalamus.

The online version of this article includes the following source data and figure supplement(s) for figure 4:

**Source data 1.** LCN2 binds to the hypothalamus of primates.

**Figure supplement 1.** LCN2 binds to the hypothalamus of primates.

**Figure supplement 1—source data 1.** LCN2 binds to the hypothalamus of primates.

*figure supplement 1, A*). Unlabeled LCN2 blocked part of the binding of labeled LCN2 (*Figure 4E*; *Figure 4—figure supplement 1, E*), indicating specific binding. Similarly, unlabeled a-MSH also blocked some LCN2 binding to the hypothalamus, indicating that LCN2 binds to MC4R (*Figure 4D*; *Figure 4—figure supplement 1, E*). That a-MSH blocked less [$^{125}$I] rh-LCN2 binding than non-radio-labeled LCN2 may suggest that, at least in primates [$^{125}$I] rh-LCN2 has a higher binding affinity for MC4R than α-MSH.

To enhance the rigor of the autoradiography experiments, we also examined LCN2 binding using immunofluorescence in baboon (*Figure 4F, G*) and human brain sections (*Figure 4H*) containing the hypothalamus. Binding was again shown in both human and baboon brain sections and quantified as the average of LCN2-positive cells (21.3 ± 1.3% and 17.6 ± 1.8%, respectively) and it was specific since it was competed by non-biotinylated LCN2 (*Figure 4G, I*). The reduction in binding was approximately 75%. If the concentration of the blocking agent is insufficient then the block may be incomplete and explain why 25% nonspecific or non-displaceable binding is observed even when the tracer and blocking drugs are almost the same. It is also possible that a slightest difference in structure may mean differences in nonspecific binding or off-target high-affinity binding (*Hamill et al., 2005*). Of note, bar graphs were obtained from a single brain section and therefore depict qualitative representations of binding. Overall, we observed a consistent and comparable degree of binding in the hypothalamus, among the three species examined (*Figure 4—figure supplement 1A–F*), which indicates that the PET findings are evidence of specific binding in the non-human primate hypothalamus and supports the premise of a common interspecies target of action for LCN2.

## rh-LCN2 treatment suppresses food intake and body weight in vervets within five days of treatment

Having established that rh-LCN2 is able to cross the BBB of vervets and localize to the hypothalamus, we then sought to examine whether a daily treatment of lean monkeys with intravenously administered rh-LCN2 would lead to appetite suppression. As described in Materials and methods, this was a cross-over study with two treatment weeks and one washout period in between (*Figure 5A*). The LCN2 dose was extrapolated from our studies in mice (*Mosialou et al., 2017*). In the mouse hypothalamus, the amount of naturally occurring LCN2 is 28 pg/mg and in the adult mouse and human serum, it is on average 100–150 ng/mL. In mice, the administration of LCN2 by intraperitoneal injection of 150 ng/g daily crosses the blood-brain barrier and suppresses appetite. Using interspecies conversion per m$^2$, we calculated the monkey dose to be 0.0375 mg/kg. This dose is equivalent to the amount used to treat mice and it is calculated based on body surface area; it takes into account the interspecies variation in several physiological parameters including oxygen utilization, caloric expenditure, basal metabolism, and blood volume (*Reagan-Shaw et al., 2008*).

Digestion of an LCN2 aliquot with N-glycanase showed that the recombinant protein is pure and N-glycosylated (*Figure 5—figure supplement 1*). In addition, each of the non-human primates received a maximum endotoxin of 0.03 EU/kg/hr, a dose well below the endotoxin limit (5 EU/kg/hr) defined as acceptable by the U.S. Food and Drug Administration (*Office of Regulatory Affairs, 2014*).

During the first treatment week, the LCN2-treated group of monkeys showed a 27% decrease in food intake compared to baseline (food intake before treatment) and a 25% decrease compared to the saline-treated monkeys (*Figure 5B*). Following the first week of treatment recombinant LCN2 was washed out by allowing the monkeys some days to recover. The following week, baseline food intake was measured (baseline 2). As shown in *Figure 5C*, both saline and LCN2 groups returned to baseline levels of food intake before the start of the second treatment week, where groups were assigned to the opposite treatment scheme (crossover). Similarly, during the second treatment week, the monkeys that received rh-LCN2 showed a 29% reduction in food intake compared to baseline and a 15% decrease compared to the saline-treated group (*Figure 5D*). When data from both treatment weeks were combined, the LCN2-treated group had a 28% reduction in food intake compared to baseline and a 21% reduction compared to the saline-treated group (*Figure 5E*). It should be noted that in *Figure 5B and D* the results show a trend toward reduction of food intake in the LCN2-treated animals; they do not reach statistical significance due to the small number of animals per group (N = 3). However, a robust and statistically significant reduction in food intake was observed when data from the two treatment weeks were combined as shown in *Figure 5E* (N = 6 animals/group).

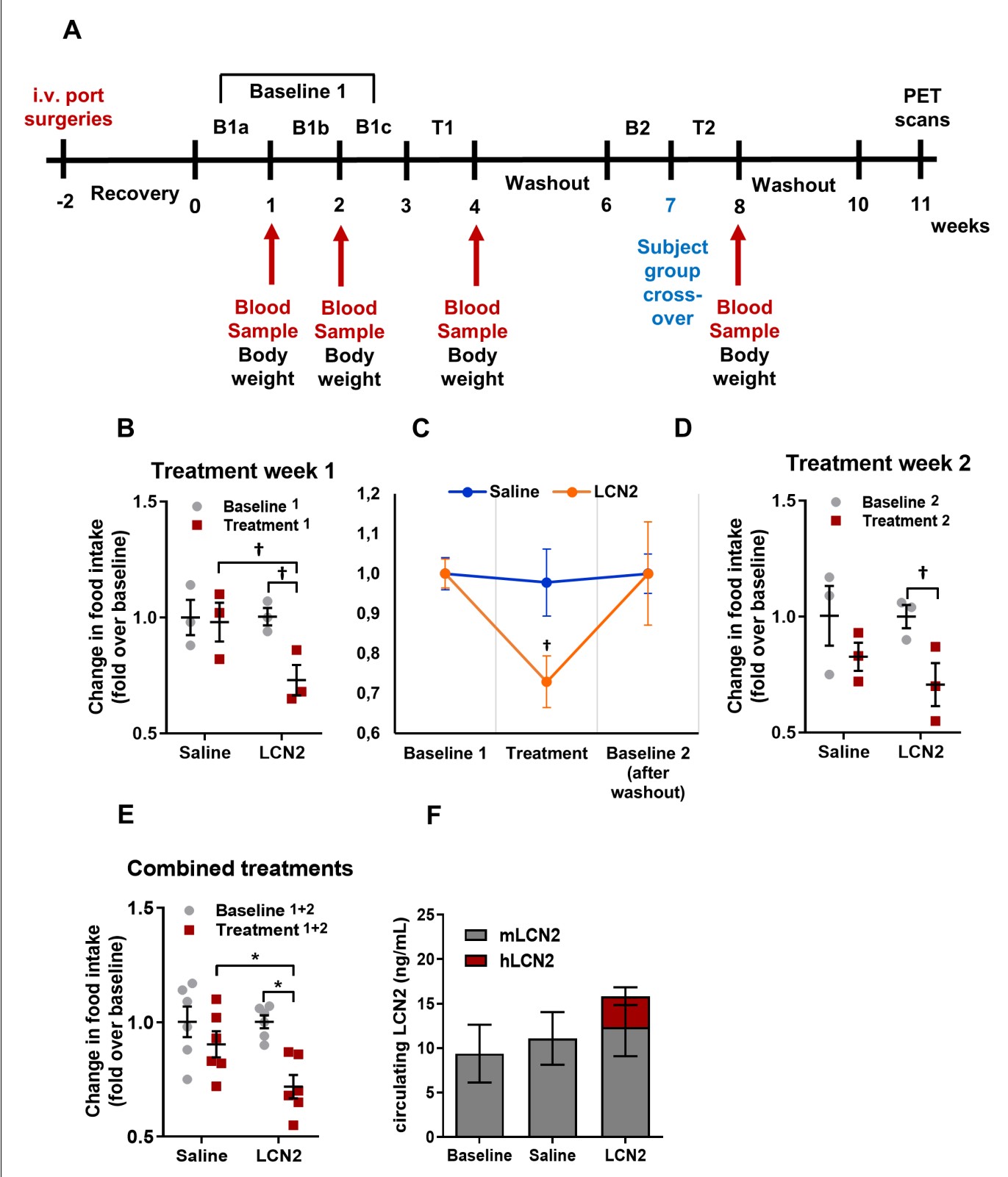

**Figure 5.** Rh-LCN2 administration suppresses food intake in vervets within 5 days of treatment. (A) Timeline in weeks showing the design and major events of the non-human primate study. (B) Change in food intake of saline- and LCN2-treated vervets during the first week of treatment (n = 3 monkeys/treatment). (C) Change in food intake of saline- and LCN2-treated vervets during the baseline, the first week of treatment, and the subsequent washout period (n = 3 monkeys/treatment). (D–E) Change in food intake of saline- and LCN2-treated vervets during (D) the second week of treatment

*Figure 5 continued on next page*

Figure 5 continued

(n = 3 monkeys/treatment) and (E) when treatment weeks were combined and values were averaged (n = 6 monkeys/treatment). (F) Circulating levels of monkey and human LCN2 in the treated monkeys. Two different ELISA assays were used; one for human and one for monkey LCN2. Each ELISA has selective reactivity for the designated species. In G, gray bars indicate serum levels of monkey whereas red bars represent human LCN2 following its administration. Values represent mean ± SEM. * indicates p<0.05 and † indicates p<0.1. B = Baseline, T = Treatment, mLCN2 = monkey Lipocalin-2, hLCN2 = human Lipocalin-2.

The online version of this article includes the following source data and figure supplement(s) for figure 5:

**Source data 1.** Rh-LCN2 administration suppresses food intake in vervets within 5 days of treatment.

**Figure supplement 1.** Rh-LCN2 is pure and N-glycosylated.

Notably, the administration of hLCN2 to monkeys led to a 1.7-fold increase in circulating levels (combined monkey and human 15.9 ng/mL) as compared to baseline levels (9.4 ng/mL) within 4 hr after injection (*Figure 5F*). It should be noted that even though the combined human and monkey LCN2 levels approximately doubled following the administration of recombinant human LCN2, this increase was not statistically significant (p=0.35), most probably due to the small number of monkeys used in each group.

To examine if the observed anorexigenic effect of LCN2 affected body weight or adiposity, we measured body weight, serum leptin, as a marker of adiposity proportional to fat stores size, and serum triglycerides, as a marker of ingested fat load and lipid metabolism. Due to the persisting anorexigenic effect of LCN2 in the second week of treatment, we focused our analysis of body weight and adiposity on the first week of treatment. We found that even during this short period of time rh-LCN2 reduced body weight (*Figure 6A, D*), serum leptin (*Figure 6B, E*), and triglyceride levels (*Figure 6C, F*) in treated animals, even though without a statistical significance, probably due to the small sample size (n = 3 per group).

## LCN2 treatment causes negligible toxicity in vervets

To determine whether the rh-LCN2 treatment causes major side effects we measured several biomarkers in the serum of the monkeys, both before and after treatment. As shown in *Table 1*, serum samples from two timepoints were collected before the first week of treatment, whose values were averaged for each subject and indicated as 'baseline 1' (n = 6 subjects). Furthermore, we collected serum samples from the end of the first and second treatment week. 'Saline' and 'LCN2' groups consisted of n = 6 subjects/serum samples, three from the first and three from the second treatment week.

First, we examined whether our recombinant protein preparation caused any acute-phase response to the treated monkeys. To test this, we measured serum CRP levels before and after the treatment. As shown in *Table 1*, while CRP levels appear to increase during the time of LCN2 injections, the effect was mostly driven by the extreme value for one animal in the first week of treatment. Rhabdomyolysis is indicated as a rare cause of CRP increase, and may account for the high levels of CRP in that subject (*Landry et al., 2017*). It is possible that the animal did not tolerate well the daily intramuscular injections of dexmetodimine, which is also evident by the AST data (the value for this animal was the highest). As we mention in the next paragraph, high levels of AST may indicate muscle inflammation or injury. Furthermore, a subject of the saline-treated group was difficult to dose throughout the study, and this stress may account for the high CRP concentrations in this subject at each time point. Nevertheless, serum CRP levels did not change with LCN2 treatment. Similarly, saline-treated monkeys had normal CRP levels compared with baseline.

Quantification of liver enzyme levels in the serum that would indicate liver injury or toxicity, showed no significant elevation with treatment, except for aspartate aminotransferase AST (SGOT). AST was elevated both in saline-treated and LCN2-treated subjects to a similar extent (*Table 1*). Higher levels of AST compared to ALT, as in this study, may indicate muscle inflammation due to myocyte injury following vigorous exercise, toxins, or drugs use (*Cobbold et al., 2010*). In this study, monkeys had to be injected daily with a mild sedative drug, dexmetodimine, which was given intramuscularly. It is possible that the daily intramuscular injections, led to myocyte injury that was in turn responsible for the rise in AST levels.

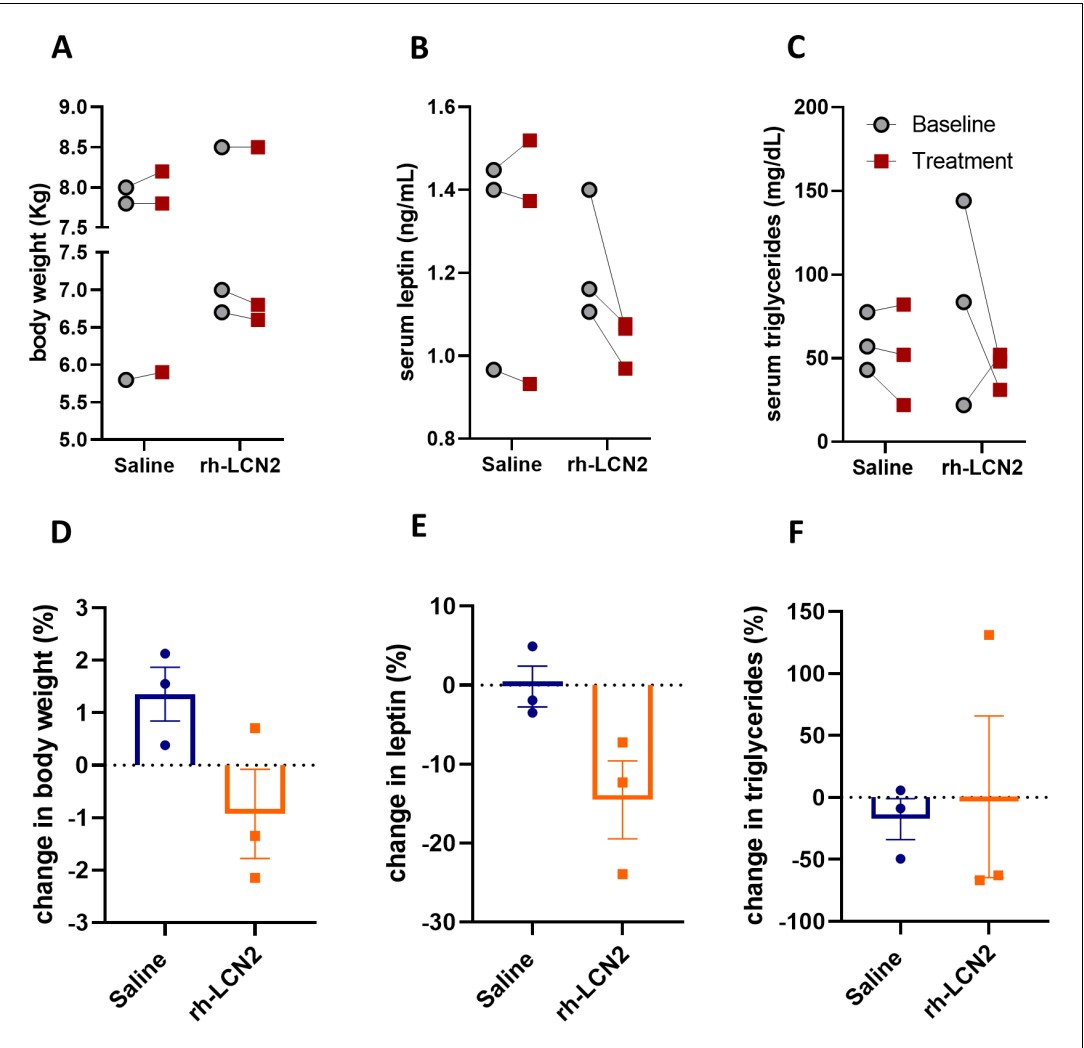

**Figure 6.** Body weight, leptin, and triglyceride show a tendency for decrease in vervet monkeys after rh-LCN2 treatment. (A) Body weight, (B) serum leptin, and (C) serum triglycerides at baseline and after saline- and LCN2-treatment of vervets (n = 3 monkeys/treatment). Change in body weight (D), serum leptin (E), and serum triglycerides (F) at the end of treatment week 1. Values represent mean ± SEM. rh-LCN2 = recombinant human Lipocalin-2.

The online version of this article includes the following source data for figure 6:

**Source data 1.** Body weight, leptin, and triglyceride show a tendency for decrease in vervet monkeys after rh-LCN2 treatment.

The kidney function and possible injury before and after treatment were assessed by measuring serum creatinine and blood urea nitrogen levels (BUN). Serum creatinine was elevated in the LCN2-treated group compared to baseline but this rise was within the normal range and did not differ significantly from the saline-treated group. Similarly, even though BUN was elevated with LCN2 treatment, this rise was again found within physiological levels and did not change significantly from the saline-treated group. In addition, the BUN/Creatinine ratio did not change with treatment ruling out substantial acute renal injury or dysfunction (*Table 1*).

Additional serum parameters, such as serum total protein, albumin, globulin, and calcium were analyzed (*Table 1*) but did not change with treatment. Serum phosphorus was elevated in the LCN2-treated monkeys but this rise was within physiological levels and did not differ significantly from the saline-treated monkeys.

**Table 1.** Acute-phase, toxicological, and metabolic markers in the treated vervets.

| Parameter | Baseline | Saline | LCN2 |
|---|---|---|---|
| Primate LCN2 (ng/mL) | 9.4 ± 3.2 | 11.1 ± 3.0 | 12.4 ± 3.3 |
| CRP (ng/mL) | 464.4 ± 253.4 | 453.1 ± 191.1 | 935.5 ± 618.7 |
| GGT (u/L) | 51.0 ± 4.5 | 52.2 ± 5.1 | 49.5 ± 4.0 |
| AST(SGOT) (u/L) | 52.8 ± 3.8 | 106.7 ± 14.8(*) | 131.3 ± 12.3(*) |
| ALT(SGPT) (u/L) | 91.7 ± 21.1 | 104.8 ± 29.3 | 93.3 ± 25.1 |
| ALP (u/L) | 100.3 ± 11.3 | 79.5 ± 14.6 | 82.5 ± 7.1 |
| BUN (mg/dL) | 13.8 ± 0.5 | 15.5 ± 0.8 | 16.3 ± 0.6(*) |
| Creatinine (mg/dL) | 0.7 ± 0.1 | 0.8 ± 0.0 | 0.8 ± 0.0(*) |
| BUN/Creatinine ratio | 19.9 ± 2.1 | 20.7 ± 0.9 | 20.5 ± 1.1 |
| Total protein (g/dL) | 6.6 ± 0.2 | 6.6 ± 0.4 | 6.7 ± 0.2 |
| Albumin (g/dL) | 4.0 ± 0.2 | 3.5 ± 0.5 | 4.1 ± 0.1 |
| Globulin (g/dL) | 2.5 ± 0.1 | 2.9 ± 0.2 | 2.6 ± 0.1 |
| Phosphorus (mg/dL) | 3.7 ± 0.2 | 4.6 ± 0.4 | 5.0 ± 0.5(*) |
| Calcium (mg/dL) | 8.6 ± 0.2 | 8.5 ± 0.3 | 8.8 ± 0.2 |
| Glucose (mg/dL) | 80.3 ± 3.3 | 64.7 ± 2.6(*) | 68.2 ± 4.4(*) |
| Insulin (mIU/mL) | 83.0 ± 16.0 | 61.5 ± 14.5 | 50.8 ± 11.8 |
| Cholesterol (mg/dL) | 141.4 ± 16.3 | 147.2 ± 20.1 | 143.8 ± 20.8 |
| Triglycerides (mg/dL) | 71.3 ± 17.2 | 51.2 ± 11.1 | 50.7 ± 11.4(†) |
| Leptin (ng/mL) | 1.3 ± 0.1 | 1.3 ± 0.1 | 1.2 ± 0.1 |
| Amylase (u/L) | 1336.8 ± 190.9 | 1699.8 ± 283.8 | 1744.5 ± 352.5 |
| Lipase (u/L) | 71.2 ± 7.2 | 85.0 ± 11.8 | 77.2 ± 11.6 |

Values represent mean ± SEM. * indicates p<0.05 and † indicates p<0.1 versus baseline, with two-tailed paired Student's t-tests. LCN2 = Lipocalin-2, CRP = C reactive protein, GGT = Gamma glutamyl transferase, AST = Aspartate transaminase, SGOT = Serum glutamate-oxaloacetate transaminase, ALT = Alanine transaminase, SGPT = Serum glutamic pyruvic transaminase, ALP = Alkaline phosphatase, BUN = Blood urea nitrogen.

The online version of this article includes the following source data for Table 1:

**Source data 1.** Acute-phase, toxicological, and metabolic markers in the treated vervets.

An examination of metabolic parameters showed a decrease in serum glucose and triglycerides and no significant changes in serum cholesterol, amylase, and lipase levels. Regarding glucose, a significant decrease was observed in both treatment groups when compared to baseline, whereas in triglycerides the reduction was borderline significant only in the LCN2-treated group. Serum insulin was not altered by LCN2 treatment. Of note, monkey LCN2 levels remained stable, both with saline and LCN2 treatment ruling out the possibility of regulation of the endogenous monkey LCN2 by exogenous administration of the recombinant human protein.

## Discussion

We examined the translational and therapeutic potential of LCN2, and we found that LCN2 crosses the BBB, localizes to the hypothalamus, and suppresses food intake in non-human primates. Consistent with this, binding experiments show the specificity of LCN2 binding in the PVN of human and non-human primates.

In addition, circulating LCN2 increases postprandially in lean human subjects, and the rise in LCN2 correlates with a drop in hunger sensation; these data suggest that the satiety function of LCN2 is conserved in humans. However, LCN2 post-prandial response is blunted in individuals with obesity. The extent of postprandial elevation of serum LCN2 response is affected by BMI, as women with normal weight have the highest response, women with overweight have a positive yet relatively

attenuated response and women with obesity have a negative response. Similar to LCN2, other hunger or satiety peptide postprandial concentrations have different responses in people with normal or excessive BMI (*Adamska-Patruno et al., 2019*). Specifically, postprandial upregulation of LCN2 is of similar magnitude as of GLP1 in individuals with normal weight or overweight and strongly correlates with hunger scores in these groups. These results suggest that the decrease in hunger sensation in humans might be mediated by the same mechanism that we observed in mice, namely LCN2 binding to appetite-suppression centers in the hypothalamus.

The observed variation in the extend of postprandial LCN2 increase among the studies could be due to differences in test meals (total mass, caloric content, macronutrient composition, and texture), the extent of involvement of the cephalic phase response and the stimulation of taste receptors by the presence of solid foods, as well as the racial composition of the cohort, all factors previously reported to affect postprandial gut peptides release and satiety responses (*Feinle-Bisset, 2014*; *Miquel-Kergoat et al., 2015*; *Willis et al., 2010*; *Lee et al., 2006*; *Cooper, 2014*; *de Graaf, 2012*). Moreover, in most studies, the magnitude of postprandial response tends to be similar for LCN2 and GLP-1, although it differs in subjects with normal or overweight. However, whereas LCN2 levels inversely correlate with hunger in normal weight and overweight subjects but not obese individuals, GLP1 levels only inversely correlate with hunger scores in subjects with obesity. Notably, the postprandial regulation of LCN2 and its role in satiety is supported by multifaceted evidence taking into consideration not only the p-value threshold, but also based on scientific reasoning in accordance to the guidelines of the American Statistical Association (*Wasserstein and Lazar, 2016*). This is important for experiments like those described in this study which involve four different cohorts of human subjects collected and evaluated at four different sites, three in the United States and one in Europe. Patients across different states and countries eat different diets, which increase the variability of the responses. The fact that within this variation range, all subjects respond by increasing LCN2 in their blood after a meal, highlights the broad significance of these observations.

While the postprandial increase in LCN2 observed in the average-weight humans was 16–50%, the one we previously observed in mice was approximately 300% within the first 3 hr (*Mosialou et al., 2017*). This difference could be due to several reasons: (1) Difference in meal patterns between the two species. Mice mainly eat in the night and overnight fasting would result in approximately 75% of food deprivation. Humans fast after midnight, which would be after dinner, and were offered a meal at or around breakfast time. Thus, fasting and hunger in humans would not be as severe as in mice and could affect the magnitude of postprandial LCN2 response after feeding. (2) Differences in the amount of food consumed within the same time after fast/refeed between mice and humans. Mice consume approximately 40% (1.6g) of their daily food intake (4 g total on average) within 3 hr. In our Study 1, 2, and 3, humans consumed 37% (732 kcal), 11% (212 kcal), and 14% (275 kcal) of their daily food intake (2000 calories/day, *U.S. Department of Health and Human Services and U.S, 2015*, Appendix 2), respectively. These numbers account for a 2.8- to 29–fold higher food intake in mice than in humans within 3 hr. Higher food intake in mice may have stimulated the higher increase in postprandial LCN2 levels.

Ours studies show that the magnitude of postprandial increase of LCN2 is inversely correlated with hunger scores in normal weight and overweight human subjects. A similar magnitude of postprandial change we observed in serum GLP-1 levels in normal weight and overweight subjects. LCN2, and GLP-1, are not the first examples of a modest increase in hormone levels, producing a large change in a biological function or in disease. For example, a 10% increase in serum levels of parathyroid hormone above normal levels is sufficient to induce hyperparathyroidism; likewise, an increase of blood calcium from 100 to 104 mg (5% increase) is sufficient to create pathological hypercalcemia. Our results demonstrate that 20–50% increase in LCN2 levels, can suppress food intake in rodents and non-human primates and inversely correlate with hunger scores in normal weight and overweight human subjects. In endocrinology, a modest increase in hormones or second messenger signaling mechanisms like those involving GPCR signaling (as in our case with LCN2 acting through the MC4R) are enough to disturb homeostasis and induce pathology.

In the future, larger and well-controlled studies are needed to more precisely characterize LCN2 as a biomarker of adverse metabolic profile and abnormal appetite control and how BMI, meal composition and texture and other subject characteristics affect the postprandial LCN2 response and subsequent appetite control. Even though more thorough pharmacological studies are warranted in

the future, this set of data suggests that LCN2 could be developed into an effective and safe treatment for obesity.

# Materials and methods

## Key resources table

| Reagent type (species) or resource | Designation | Source or reference | Identifiers | Additional information |
|---|---|---|---|---|
| Antibody | Rabbit anti-biotin antibody | Abcam | Cat# ab53494, RRID:AB_867860 | 10 ug/mL |
| Antibody | Donkey anti-rabbit Alexa Fluor 488 | Life technologies | Cat# A-21206, RRID:AB_2535792 | 1:200 |
| Biological sample (*Homo sapiens*) | RNA from human osteoblasts | Laboratory of Prof. Stavroula Kousteni | | The coding sequence of human LCN2 was amplified and the amplified insert was cloned into the πα-SHP-H vector. Additional information can be found in section: 'Production of recombinant human LCN2 (rh-LCN2)". |
| Biological sample (*Homo sapiens*, Female) | Study 1 serum samples | Columbia University Irving Medical Center | | Additional information can be found in section: 'Subjects, protocols and test meals'. |
| Biological sample (*Homo sapiens*, Female) | Study 2 serum samples | University of Lyon - INSERM UMR 1033 institute | | Additional information can be found in section: 'Subjects, protocols and test meals'. |
| Biological sample (*Homo sapiens*, Male and Female) | Study 3 serum samples | Rutgers University | | Additional information can be found in section: 'Subjects, protocols and test meals'. |
| Biological sample (*Homo sapiens*, Male and Female) | Study 4 plasma samples | Columbia University Irving Medical Center | | Additional information can be found in section: 'Subjects, protocols and test meals'. |
| Biological sample (*Chlorocebus aethiops sabeus*, Male) | African Green Monkeys serum samples | Wake Forest School of Medicine | | Additional information can be found in section: 'Non-human primate study'. |
| Biological sample (*Homo sapiens*, Male) | Post-mortem human brain sections | Columbia University – New York State Psychiatric Institute | | Additional information can be found in section: 'Autoradiography and immunofluorescence on brain sections from primates'. |
| Biological sample (*Papio anubis*, Male) | Baboon brain sections | Columbia University – New York State Psychiatric Institute | | Additional information can be found in section: 'Autoradiography and immunofluorescence on brain sections from primates'. |
| Biological sample (*Macaca fascicularis*, Male) | Macaque brain sections | Columbia University – New York State Psychiatric Institute | | Additional information can be found in section: 'Autoradiography and immunofluorescence on brain sections from primates'. |
| Chemical compound, drug | PEI max | Polysciences Inc | Cat# 24765–1 | |
| Chemical compound, drug | Radioactive sodium iodide (Na$^{125}$I) | Perkin Elmer | Cat# NEZ033001MC | |
| Peptide, recombinant protein | Alpha MSH | Tocris | Cat# 2584 | |
| Peptide, recombinant protein | Recombinant Human LCN2 (rh-LCN2) | This paper | | Additional information can be found in section 'Production of recombinant human LCN2 (rh-LCN2)". |
| Peptide, recombinant protein | N-glycanase | Sigma-Aldrich | Cat# P9120 | 0.1 U |

*Continued on next page*

*Continued*

| Reagent type (species) or resource | Designation | Source or reference | Identifiers | Additional information |
|---|---|---|---|---|
| Peptide, recombinant protein | NHS Biotin | Thermo Fisher Scientific | Cat# 20217 | Additional information can be found in section: 'Autoradiography and immunofluorescence on brain sections from primates'. |
| Commercial assay or kit | Human Lipocalin-2/NGAL DuoSet ELISA | R and D Systems | Cat# DY1757 | |
| Commercial assay or kit | Human Insulin ELISA | Crystal Chem | Cat# 90095 | |
| Commercial assay or kit | Human GLP-1 ELISA | Merck Millipore | Cat# EZGLP1T-36K | |
| Commercial assay or kit | Monkey Lipocalin-2 ELISA | LifeSpan BioSciences | Cat# LS-F38530 | |
| Commercial assay or kit | Monkey Insulin ELISA | LifeSpan BioSciences | Cat# LS-F10306 | |
| Commercial assay or kit | Primate CRP | Helica Biosystems | Cat# 911CRP01P-96 | |
| Commercial assay or kit | Monkey Leptin | Cusabio | Cat# CSB-E14936Mk | |
| Other | Raw data (Human studies 1–4) | This paper | | Raw data can be found in the source data files 'Figure 1—source data 1', 'Figure 1—figure supplement 1—source data 1'. |
| Other | Raw data (Non-human primate study) | This paper | | Raw data can be found in the source data files of Figures 3–6. |
| Other | Raw data (Autoradiography studies) | This paper | | Raw data can be found in the source data files 'Figure 4—source data 1' and 'Figure 4—figure supplement 1—source data 1' |
| Cell line (Homo-sapiens, Female) | HEK-293 | ATCC | Cat# CRL-1573, RRID:CVCL_0045 | |
| Cell line (Homo-sapiens, Female) | Expi293 | ThermoFisher Scientific | Cat# A14527 RRID:CVCL_D615 | |
| Recombinant DNA reagent | Plasmid: πα-SHP-H-LCN2 | This paper | | Additional information can be found in section 'Production of recombinant human LCN2 (rh-LCN2)". |
| Software, algorithm | GraphPad Prism v8 | GraphPad | RRID:SCR_002798 | https://www.graphpad.com/scientific-software/prism/ |
| Software, algorithm | SAS v9.4 | SAS | RRID:SCR_008567 | https://www.sas.com |
| Software, algorithm | Pmod software | Pmod Technologies | RRID:SCR_016547 | https://www.pmod.com/web/ |

## Human studies

Data from four separate studies were used to assess the relationship between LCN2 levels and postprandial regulation in healthy, overweight, obese, and severely obese humans. Study 1 was performed at the Department of Medicine-Endocrinology of Columbia University Medical Center (CUMC), Study 2 at the INSERM UMR 1033 institute of the University of Lyon, Study 3 at the Department of Nutritional Sciences of Rutgers University and Study 4 at New York Obesity Nutrition Research Center of CUMC.

## Subjects, protocols, and test meals

### Study 1

This study was designed to examine whether the ingestion of a liquid meal by healthy normal-weight women after an overnight fast has any effect on serum LCN2 concentrations and whether this correlates with hunger reduction. Eleven young women (mean age ± SEM: 27.9 ± 0.7 years) with normal body weight (BMI ± SEM: 21.8 ± 0.6 kg/m$^2$) participated in the study.

A total of 63 participants were screened and 11 participants were eligible for recruitment by meeting the inclusion/exclusion criteria. Exclusion criteria included history of diabetes, untreated hyperthyroidism/hypothyroidism, cancer, cardiovascular disease, inflammatory diseases, past surgery, or medications that would affect appetite, ingestion, digestion, absorption, or metabolism of nutrients, weight loss of >5% of body weight in the past 3 months, excessive caffeine use (>6 caffeinated beverages/day) and current smoking. The characteristics of 11 participants (63.6% Caucasian, 36.4% Asian) are summarized in *Supplementary file 1A*.

All visits took place between days 3 through 9 of the follicular phase of the menstrual cycle. Subjects received a liquid meal (732 kcal: 65% fat, 5% protein, 30% carbohydrates) at 8:00 am following an overnight fast (no food or drink excluding water for 8–12 hr). Participants had 15 min to drink all of the meal. A visual analog score (VAS) was used to assess hunger sensation before and after the ingestion of the meal. The scale has a 100 mm horizontal line with the most positive and the most negative rating at each end. Participants chose a point on the line representative of their current perception of hunger. The distance in mm between the leftmost (zero) point and the point marked by the participant was measured and used to determine the VAS score. All of the participants successfully completed the liquid-meal tolerance test. LCN2, glucose, and insulin serum concentrations were measured before and after ingestion of the meal.

### Study 2

This study was conducted in Lyon, France and it has been designed to investigate whether LCN2 levels change after a meal challenge in a small cohort of young normal-weight women. Volunteers were eligible if they were ≥18 years old, had a body mass index (BMI) between 18 and 25 kg/m$^2$, performed less than 120 min of physical activity per week, and had a normal rest electrocardiogram. Participants were excluded if they were pregnant, had any cardiovascular risk factor or had any chronic disease including diabetes history and hypertension. Participants were informed about the protocol and provided a signed informed. Participants were given a mixed breakfast-type meal (212 kcal: 34% fat, 8% protein, 58% carbs) following an overnight fast. LCN2 serum levels were measured before and after the ingestion of the test meal. Participants remained seated at rest during the experiment. This study included nine young (mean age ± SEM: 26.1 ± 1.0 years) normal-weight women (mean BMI ± SEM: 20.8 ± 0.5 Kg/m$^2$) whose characteristics are presented in *Supplementary file 1A*.

### Study 3

With this study, we sought to address whether postprandial regulation of LCN2 is conserved in overweight and obese people. To this end, 47 overweight or obese individuals (mean age ± SEM: 32.4 ± 1.8 years) who were otherwise healthy, with a BMI between 25–35 kg/m$^2$ were recruited.

Individuals were excluded if there was a diagnosis of an eating disorder, gastrointestinal illness, bariatric surgery, hyperparathyroidism, untreated thyroid disease, diabetes, blood pressure >140/90, significant immune, hepatic, or renal disease, significant cardiac disease, active malignancy, or cancer therapy within the past year, current use of obesity medications or dietary supplements or any weight regimen. The trial is registered at clinicaltrials.gov (NCT02929849).

After an overnight fast (no food intake after 9 pm) body weight, blood pressure, body fat, and waist circumference were measured and a baseline fasting (0 min) blood sample drawn. Participants also completed multiple visual analog scales (VAS) before breakfast to measure their subjective appetite sensations (hunger, fullness, prospective food consumption, and satiety). After baseline blood withdrawal and VAS evaluation, each participant was served a breakfast meal (275 kcal; 50% carbohydrate; 30% fat; 20% protein). Afterward, postprandial blood was drawn and VAS measurements were taken over a 3 hr period. Some data from this study has been published elsewhere (*Hao et al., 2017*).

## Study 4

Here we sought to address whether postprandial regulation of LCN2 is restored in individuals with severe obesity after a Roux-en-Y gastric bypass (RYGB). Fasting and postprandial LCN2 concentrations were measured for 4 hr after a liquid test meal ingested after an overnight fast by 12 individuals (11 women, 1 man) with severe obesity and without type 2 diabetes (mean age ± SEM: 39.3 ± 5.3 years, BMI = 45.8 ± 3.0 kg/m$^2$), before and again one year after Roux-en-Y gastric bypass surgery. The timing of meal ingestion was controlled for in all subjects. A visual analog score (VAS) of 150 mm was used to assess hunger before and after the ingestion of the meal. Details of the experimental protocol and some data from this study have been published elsewhere (*Stano et al., 2017*). The study was registered at clinicaltrials.gov (NCT02929212).

### Human blood sampling

An intravenous catheter was placed in the antecubital vein and fasting-baseline blood sample was taken (t = 0). A normal saline (0.9%) flush was used after each blood draw. Following ingestion of the test meal, blood samples were taken at 15, 30, 45, 60, 90, 120, and 180 min for Study 1, at 60 and 105 min for Study 2, at 30, 60, 90, 120, and 180 min for Study 3 and at 15, 30, 45, 60, 90, 120, 180, 240, 300, and 360 min for Study 4. Each blood draw consisted of a total of 10 mL of blood and was collected into serum-separator tubes (Becton Dickinson, Franklin Lakes, NJ). Blood samples were left to clot at room temperature for 30 min and were subsequently spun at 3400 rpm for 15 min at 4°C. In Study 4, blood was collected in chilled EDTA tubes with aprotinin (500 kallikrein inhibitory units/mL of blood) and dipeptidyl peptidase-4 inhibitor (10 µL/mL of blood; Millipore, Burlington, MA). Serum or plasma was then aliquoted and stored at −80°C in cryovials until assayed.

### Production of recombinant human LCN2 (rh-LCN2)

Total RNA from human osteoblasts was extracted using standard protocols. The coding sequence of human LCN2 was amplified and the amplified insert was cloned into the πα-SHP-H vector, kindly provided by Prof. Shapiro. The correct clone was verified by sequencing. Before proceeding to a full-scale production of the protein, we tested the expression in HEK-293 (ATCC) cells by qRT-PCR.

To express the rh-LCN2, 1.56 mL of PEI max transfection reagent (Polysciences Inc, Warrington, PA) was added to 25 mL Opti-MEM medium (Life Technologies, Carlsbad, CA) and incubated for 5 min at room temperature (RT). Meanwhile, 500 µg of the πα-SHP-H-LCN2 vector was added to 25 mL of Opti-MEM medium in another tube. The 25 mL Opti-MEM medium containing 1.56 mL of PEI max was then mixed with 25 mL Opti-MEM medium containing 500 µg of the πα-SHP-H-LCN2 plasmid, incubated for 15 min at RT, and added to 800 mL of Expi293 cells (Life Technologies,) at 0.9–1 million cells/mL. The transfected cells were cultured in a shaker incubator at 120 rpm, 37°C, 9% $CO_2$ overnight. The cell lines used for these assays were purchased from commercial sources and were free of mycoplasma contamination.

Four days after transfection, supernatants were harvested and purified over 13 mL nickel-charged resin (Ni-NTA Agarose; QIAGEN, Venlo, the Netherlands) in columns. Isolated rh-LCN2 was eluted with imidazole buffer (500 mM NaCl, 10 mM Tris pH = 8.0, 250 mM imidazole). Rh-LCN2 was then buffer exchanged in aqueous buffer, pH = 8.0 (150 mM NaCl, 10 mM Tris pH = 8.0) by dialysis and adjusted to a concentration of 1 mg/mL and filtered (0.22 µm).

The purity and integrity of recombinant human LCN2 stocks was assessed by SDS–PAGE followed by Coomassie blue staining. To verify proper N-glycosylation of the protein, 15 ug of the protein were digested with N-glycanase (Sigma-Aldrich, St. Louis, MO) for 16 hr at 37°C, according to the manufacturer's instructions. In addition, in the same stocks, we measured endotoxin levels by a chromogenic LAL assay (Genscript, Piscataway, NJ). We then used an endotoxin removal kit (Genscript) to further lower the endotoxin levels.

### Radiosynthesis of [$^{125}$I] rh-LCN2

The radiolabeling of rh-LCN2 was achieved by following a modified protocol by Dong (*Dong et al., 2002*). Rh-LCN2 (200 µL, 1.0 mg/mL solution) was added to a yellow-capped iodogen tube (Pierce Iodination Tube; Thermo Fisher Scientific, Waltham, MA). This was then diluted in 100 µL of Dulbecco's phosphate buffer saline (DPBS, 1×, with calcium chloride and magnesium chloride, pH 7.0–7.2; Gibco-Thermo Fisher Scientific) to bring to a calculated 0.75 mg/mL concentration. Twenty-five

microliters (25 µL) of radioactive sodium iodide (Na$^{125}$I in 0.1 M NaOH; Perkin Elmer, Waltham, MA) was added to this solution (assay 635 µCi). The reaction mixture was sealed and incubated for 60 min at room temperature. After 60 min, the reaction mixture was passed through a preconditioned (with 100 mL of DPBS, 1× solution) size-exclusion column (PD-10 Sephadex G-25M column, Part No. 17085101OL/AG; GE Healthcare, Chicago, IL) with the use of DPBS. Fractions were collected in 1.0 mL increments. The resulting eluent (~2 mL) was assayed to give 150 µCi (24% rcy) of [$^{125}$I]-I-rh-LCN2. TLC: 1 µL of [$^{125}$I]-I-rh-LCN2 sample was spot on the iTLC-SG plate and developed in 70:30 MeOH: H2O. 'Free' (i.e. unlabeled) iodine moved with the solvent up the iTLC-SG plate, while the desired I-125 labeled complex stuck to the baseline. The strip was then cut into thirds, placed in tubes for gamma counter analysis (Hidex Automatic Gamma Counter; Hidex, Turku, Finland) and run on an open window.

## Autoradiography and immunofluorescence on brain sections from primates

### Autoradiography

Postmortem human brain sections (20 µm) from non-psychiatric controls (n = 3) that included the hippocampus, hypothalamus and thalamus were used. Baboon (*Papio anubis*, n = 1) and rhesus macaque monkey (*Macaca fascicularis*, n = 3) brain sections (20 µm) from midcallosal levels were used for *in vitro* studies. Tissue sections were pre-incubated in Tris buffer (pH 7.4) containing 0.1% BSA for 30 min. Tissues sections were then added to a solution of Tris/BSA buffer solution with 200pM [$^{125}$I] rh-LCN2 (~1.50 µCi/mL) for 60 min at RT. Adjacent sections were incubated with 1 µM rh-LCN2 or 1 µM α-MSH to determine non-specific binding. Sections were washed for 15 min (1 × 10 min followed by 1 × 5 min) in Tris/BSA buffer solution at 4°C followed by a dip in cold distilled H$_2$O to remove buffer salts. Sections were then dried under a stream of desiccated cold air. All slides were laid out in X-ray film cassettes and exposed to Biomax MR film (Kodak, Rochester, NY) for 1–2 days, developed using Kodak D-19 developer and fixative. The autoradiograms were sampled using a computer-based image analysis system (MCID; Imaging Research Inc, Ontario, Canada) as previously described (*Arango et al., 1995*; *Arango et al., 1993*; *Boldrini et al., 2008*). All the sections were corrected for light transmission inhomogeneities and binding was quantified by relative optical density (ROD, a grayscale which does allow for scaling of the amount of darkness in the film).

### Immunofluorescence

Human and baboon brain sections were rehydrated in ice-cold binding buffer (50 nM Tris-HCl [pH 7.4], 10 nM MgCl$_2$, 0.1 mM EDTA, and 0.1% BSA) for 15 min and incubated 1 hr at room temperature in the presence of biotinylated LCN2 (25 pg/mL−1). After washing in harvesting buffer (50 mM Tris-HCl [pH 7.4]), samples were fixed in 4% PFA for 15 min, washed in PBS, and incubated with rabbit anti-biotin antibody (ab53494, Abcam, Cambridge, UK) overnight at 4°C. The signal was visualized, after incubation with anti-rabbit Alexa Fluor 488 (A21206, Life Technologies, Carlsbad, CA) followed by DAPI counterstaining. To test for assay specificity, the procedure described above was performed in the presence of hundred-fold excess of non-biotinylated LCN2 (2.5 ng/mL). The binding was quantitated using the 'Cell Counter' analysis in ImageJ, by counting total cells (DAPI-stained/blue) and LCN2-positive cells (LCN2-stained/green) and subsequently calculating the percent of LCN2-positive cells in the two conditions.

## Non-human primate study

Based on our previous mouse studies where we observed an 18% reduction in food intake in wild-type lean mice that were treated with recombinant LCN2, we calculated that in order to have an 18% reduction (SD=+/-5 g), and error a = 0.05 and power of analysis 0.90, we needed n = 6 monkeys per group (*Brant, 2020*). To minimize the number of animals we employed a cross-over design, in which each animal served as its control. There were two treatment weeks with one washout period of 9 days in between. In the first treatment week, three monkeys were treated with saline and three with rh-LCN2. After the washout period monkeys were assigned to the opposite treatment group (*Figure 4A*).

Six male vervet/African green monkeys (AGM; Caribbean-origin *Chlorocebus aethiops sabaeus*) used in this study were housed at the Vervet Research Colony at Wake Forest School of Medicine. The age of the subjects ranged from 4.7 to 16.6 years. Vervets were pair-housed in standard four cage racks (81 × 71 × 81 cm per compartment) that contained sitting perches and enrichment items. Animals had unlimited access to commercial monkey chow (LabDiet 5038; Labdiet, St Louis, MO) and water via automated lixit. They were fed supplemental fruits and vegetables 5 days/week.

Vervets were initially anesthetized with ketamine (15 mg/kg, intramuscularly-i.m.), intubated, and maintained on a combination of sevoflurane (2.0–4.0%) and isoflurane (1.5–2.0%). A catheter was surgically placed in the right saphenous vein with a vascular access port (Swirl Phantom P-SPH; Access Technologies, Skokie, IL) located subcutaneously in the right thigh. Monkeys were allowed to recover for 14 days. After recovery, they were re-acclimatized to pair-housing and food intake measurements for 7 days. At the end of the acclimatization period, the baseline food consumption for 5 days was measured. Animals were injected with rh-LCN2 or saline in the morning and then pairs were separated and fed individually for 4 hr. After the 4 hr feeding periods, all food was removed and weighed and then pairs were reunited. This means that animals were essentially fasted each night. Some enrichment foods (small pieces of fruit or vegetables) were provided in the afternoon, but those accounted to be a low percentage of their daily caloric intake. On blood sampling days, the animals had samples collected in the afternoon, after the 4 hr feeding period. Three animals were injected daily, via vascular access port, with saline and three with rh-LCN2 for five 5 days. To facilitate daily IV injections via the vascular access port, animals were lightly sedated with dexmedetomidine (0.035 mg/kg, i.m.) and then administered reversal agent atipamezole (0.35 mg/kg) following access port injections. For afternoon blood samples at the end of the week, they were anesthetized with ketamine.

Food consumption was measured daily as described above. At the end of day 5, a blood sample was collected for analyses, under ketamine anesthesia (10 mg/kg). A washout period of 9 days followed and then monkeys were subjected to a second baseline food consumption for 5 days. Then animals were switched to the opposite group. Again three animals were injected daily with saline and three with rh-LCN2 for 5 days. Food consumption was measured daily as described above. At the end of day 5, a small blood sample was collected for analyses. After the end of the second treatment week, monkeys were left to washout for 10 days. Two out of the six monkeys underwent PET imaging with a no-chase or chase design, as described below. At the end of all the studies, monkeys were anesthetized as described before and vascular access ports were surgically removed and the animals were left to recover for 14 days.

## PET imaging of [$^{124}$I] rh-LCN2 in vervets

PET scans were performed in two of the six vervets using a GE 64-slice PET/CT Discovery VCT scanner (General Electric). The day prior to the PET scan, animals were administered a solution (0.5 mL) containing 4% potassium iodide and 2% iodine via oral gavage under ketamine anesthesia (10 mg/kg, i.m.). For each scan, the fasted animal was initially immobilized with ketamine (15 mg/kg, i.m.) and maintained on 1.5–2.0% isoflurane anesthesia via an endotracheal tube. The core temperature was kept constant at 37°C with a heated water blanket. An intravenous infusion line with 0.9% NaCl was maintained during the experiment and used for hydration and radiotracer injection. The head was positioned at the center of the field of view, and a 10 min transmission scan was performed before the tracer injection. [$^{124}$I] rh-LCN2 radiotracer was injected (407 ± 80 MBq) as an intravenous bolus over 30 s, and emission data were collected for 120 min in three-dimensional mode.

In a subsequent chase experiment, α-MSH—ligand of the MC4 receptor—was given 15 min after the administration of [$^{124}$I] rh-LCN2 to block non-specific binding in the areas of interest. The same procedure, software and atlas, as in the no-chase experiment, were used to extract time-activity curves (TACs) in the thalamus and hypothalamus, bilaterally. The TACs from the chase experiment were then compared to the corresponding TACs from the first, no-chase experiment in the same regions. We compared the average standard uptake value (SUV) between the two experiments, using the formula: 100*[SUV (no-chase) − SUV (chase)]/SUV (no-chase), to calculate the percent difference. Animals had MRI scans performed 1 week before PET scans using similar anesthesia procedures. MRI scans were performed using a Siemens MAGNETOM Skyra 3T MRI Scanner (Siemens, Munich, Germany) with the following sequences: large fov localizer, small fov localizer 128 mm, MPRAGE Axial_ND, and MPRAGE Axial (TE = 3.39 ms, TR = 2700 ms, TI = 880 ms, resolution = 0.5

× 0.5 × 0.5 mm; FOV = 128 mm × 128 mm; matrix size = 256 × 256). All PET images were coregistered to the corresponding MRI (see volume of interest (VOIs) and outcome variable of SUV on PET).

## PET data analysis reconstruction

PET scans were reconstructed using the iterative ordered-subset expectation-maximization (OSEM) algorithm, which were corrected for attenuation, scatter, and dead time. The radioactivity was corrected for physical decay to the injection time and rebinned to 23 dynamic PET frames of 256 (left-to-right) by 256 (nasion-to-inion) by 000 (neck-to-cranium) voxels. The frame schedules were two 30 s, three 60 s, five 120 s, four 240 s, and nine 600 s for 120 mins. The final spatial resolution is expected to be less than 00-mm-full width half maximum in three directions.

## Volume of interest (VOIs) and outcome variable of SUV on PET

For spatial normalization, tissue segmentation, and anatomical labeling, PMod software (version 3.7; PMod Technologies Ltd, Zürich, Switzerland) was used with INIA 19 rhesus high-quality template for non-human primate brains (Rohlfing T, 2012). INIA 19 template was created from high-resolution, T1-weighted magnetic resonance (MR) images of 19 rhesus macaque (*Macaca mulatta*) animals (*Collaboratory, 2020*), including > 100 brain regions per side and transferred to individual animal's magnetic resonance imaging (MRI) using MRI-to-MRI spatial normalization (target: INIA 19 monkey MRI). Then VOIs were transferred to the PET scan of the animal using the MRI-to-PET co-registration module, and minimally adjusted for radioactivity distribution of the scan. Averaged SUV images and time-activity curves (TACs) were generated.

## Quantitation of circulating factors

Human serum LCN2, insulin, and GLP-1 were quantitated using commercially available ELISA kits and following the manufacturer's instructions (#DY1757, R and D Systems, Minneapolis, MN;#90095, CrystalChem, Inc, Elk Grove Village, IL,; #EZGLP1T-36K, Merck, Burlington, MA, respectively). Monkey LCN2 and insulin were assayed with ELISA kits from LifeSpan BioSciences (#LS-F38530 and LS-F10306 respectively; Lifespan Biosciences, Inc, Seattle, WA) and leptin with an ELISA kit from Cusabio (#CSB-E14936Mk, Cusabio, Houston, TX). Circulating levels of primate CRP were determined using a commercially available ELISA kit using the manufacturer's instructions (911CRP01P-96; Helica Biosystems, Inc, Santa Ana, CA). Blood chemistry was performed using a Heska Element DC Veterinary Chemistry Analyzer (Heska, Loveland, CO) at Columbia University's Institute of Comparative Medicine Diagnostic Laboratory.

## Statistical analysis

Analyses were performed with SAS 9.4 (SAS Institute, Inc, Cary, NC) and GraphPad Prism 8 (GraphPad, San Diego, CA), with a level of significance at $p < 0.05$. Since serum LCN2 concentration for each sample was measured multiple times over time, a one- or two-way Analysis of Variance (ANOVA) repeated measures design was appropriate for examining whether statistically significant differences in means exist in each experiment. Data were examined for normality using the Shapiro-Wilk test. Variables were log-transformed. Variables were log-transformed using the natural logarithm (ln) when appropriate, and nonparametric tests were used when necessary. Using the F-ratio statistic, we examined whether the repeated measured variable, time, had an overall statistically significant effect, implying that the corresponding values of the outcome that is examined in each experiment is statistically different from values obtained at other points of time. To correct for any violations of the assumption of sphericity ($\epsilon$), degrees of freedom for each F value were adjusted according to the estimated epsilon obtained in each analysis. For values of $\epsilon < 0.75$, the Greenhouse-Geisser Epsilon correction was used, while in cases when $\epsilon > 0.75$, the less conservative Huynh-Feldt Epsilon correction was preferred. Using the CONTRAST option in PROC General Linear Model (GLM) in SAS, we investigated whether and which mean values at each time point were statistically different from baseline, at t = 0. Pearson's correlation coefficient, or Spearman's correlation for data not normally distributed, were used. For non-repeated observations either two-tailed Student's t-tests, Mann-Whitney or rank-sum Wilcoxon non-parametric tests were used depending on the normality of distribution. For the vervet studies, comparisons were made with paired or unpaired

t-tests as needed. Data are reported as mean ± SEM. */# indicates p<0.05, ‡ indicates p<0.06, and † indicate p<0.1. n indicates the number of human subjects or animals tested in each experiment.

## Acknowledgements

Human Study #1 and the non-human primate study were supported by the National Center for Advancing Translational Sciences, National Institutes of Health, through Grant Number UL1TR001873 to Stavroula Kousteni and Mishaela Rubin. Research reported in this publication was supported by the National Heart, Lung, And Blood Institute of the National Institutes of Health under Award Number T32HL120826. Additional support was provided from the National Institute on Aging through grant number P01AG032959 awarded to Stavroula Kousteni. The content of this publication is solely the responsibility of the authors and does not necessarily represent the official views of the NIH. Human Study #2 was supported by a research grant from Roche-Chugai to Cyrille Confavreux. Human Study #3 was supported by OmniActive Health Technologies and USDA-NIFA hmg (NJAES - 0153866). Human Study #4 was supported by grants from the NIH (R01 DK067561, P30 DK26687-30, and P30 DK063608) and by the National Center for Advancing Translational Sciences, National Institutes of Health, through Grant Number UL1 TR000040.

Additional support for the vervet/African green monkey colony was provided by P40-OD010965 and UL1-TR001420. We thank Dr. Heather DeLoid for performing vascular access port surgeries and Stacey Combs, Kortne Hudick, Eric McCloud, Justin Herr, Edison Floyd, and Stephanie Rideout Danner for technical assistance.

We also thank Columbia University's Institute of Comparative Medicine Diagnostic Laboratory for blood chemistry analyses of the non-human primate study. Furthermore, we would like to thank Dr. Franck Oury for helpful discussions on the PET study.

## Additional information

### Funding

| Funder | Grant reference number | Author |
| --- | --- | --- |
| National Institutes of Health | UL1TR001873 | Mishaela Rubin<br>Stavroula Kousteni |
| National Institute on Aging | P01AG032959 | Stavroula Kousteni |
| Roche-Chugai | | Cyrille B Confavreux |
| National Institute of Food and Agriculture | NJAES - 0153866 | Sue Shapses |
| National Institutes of Health | P30DK063608 | Blandine Laferrère |
| National Institutes of Health | RO1DK067561 | Blandine Laferrère |
| National Institutes of Health | T32HL120826 | Stavroula Kousteni |
| National Institutes of Health | P30DK26687-30 | Blandine Laferrère |
| National Institutes of Health | UL1TR000040 | Blandine Laferrère |
| National Institutes of Health | UL1TR001420 | Matthew J Jorgensen |
| National Institutes of Health | P40OD010965 | Matthew J Jorgensen |

The funders had no role in study design, data collection and interpretation, or the decision to submit the work for publication.

### Author contributions

Peristera-Ioanna Petropoulou, Ioanna Mosialou, Conceptualization, Resources, Data curation, Formal analysis, Supervision, Validation, Investigation, Visualization, Methodology, Writing - original draft, Project administration, Writing - review and editing; Steven Shikhel, Conceptualization, Resources, Data curation, Validation, Investigation, Methodology, Writing - review and editing; Lihong Hao, Konstantinos Panitsas, Resources, Data curation, Formal analysis, Investigation, Writing - review and

editing; Brygida Bisikirska, Data curation, Formal analysis, Investigation, Writing - review and editing; Na Luo, Formal analysis, Investigation, Writing - review and editing; Fabiana Bahna, Resources, Investigation, Writing - review and editing; Jongho Kim, Resources, Data curation, Software, Validation, Investigation, Visualization, Writing - review and editing; Patrick Carberry, Resources, Data curation, Software, Validation, Investigation, Visualization, Methodology, Writing - review and editing; Francesca Zanderigo, Norman Simpson, Resources, Data curation, Software, Formal analysis, Validation, Investigation, Visualization, Methodology, Writing - review and editing; Mihran Bakalian, Suham Kassir, Lawrence Shapiro, Kiran Kumar Soligapuram Sai, Resources, Investigation, Methodology, Writing - review and editing; Mark D Underwood, Christina M May, Resources, Data curation, Formal analysis, Validation, Investigation, Visualization, Methodology, Writing - review and editing; Matthew J Jorgensen, Conceptualization, Resources, Formal analysis, Supervision, Validation, Investigation, Methodology, Project administration, Writing - review and editing; Cyrille B Confavreux, Conceptualization, Resources, Formal analysis, Supervision, Funding acquisition, Validation, Investigation, Methodology, Project administration, Writing - review and editing; Sue Shapses, Resources, Supervision, Funding acquisition, Investigation, Writing - review and editing; Blandine Laferrère, Resources, Data curation, Supervision, Funding acquisition, Writing - review and editing; Akiva Mintz, Conceptualization, Resources, Data curation, Supervision, Funding acquisition, Validation, Methodology, Writing - review and editing; J John Mann, Conceptualization, Resources, Data curation, Supervision, Validation, Methodology, Writing - review and editing; Mishaela Rubin, Conceptualization, Resources, Data curation, Supervision, Funding acquisition, Validation, Methodology, Project administration, Writing - review and editing; Stavroula Kousteni, Conceptualization, Resources, Data curation, Supervision, Funding acquisition, Validation, Investigation, Visualization, Methodology, Writing - original draft, Project administration, Writing - review and editing

### Author ORCIDs

Peristera-Ioanna Petropoulou (ID) https://orcid.org/0000-0001-7844-3010
Ioanna Mosialou (ID) https://orcid.org/0000-0003-0339-3316
Lihong Hao (ID) http://orcid.org/0000-0001-7795-6981
Mihran Bakalian (ID) http://orcid.org/0000-0002-1935-5982
Lawrence Shapiro (ID) http://orcid.org/0000-0001-9943-8819
Matthew J Jorgensen (ID) http://orcid.org/0000-0002-0977-6425
Stavroula Kousteni (ID) https://orcid.org/0000-0001-5163-3551

### Ethics

Human subjects: The Institutional Review Boards of Columbia University (protocol numbers for human studies 1 and 4 respectively: IRB-AAAQ9254 and IRB-AAA03051), University of Lyon (Comité de Protection des Personnes SUD-EST IV - France, protocol number for human study 2: RCB-2014-A00237-40) and Rutgers University (protocol number for human study 3: IRB-15-524M) approved these studies and informed written consent and consent to publish was obtained from each subject. All samples were obtained in accordance with the Health Insurance Portability and Accountability Act (HIPAA). All clinical investigation has been conducted according to Declaration of Helsinki principles.

Animal experimentation: Animal studies were performed in strict accordance with the recommendations in the Guide for the Care and Use of Laboratory Animals of the National Institutes of Health. All of the animals were handled according to the approved institutional animal care and use committee (IACUC) protocol (#A18-067) of Wake Forest School of Medicine. All surgery was performed under sodium pentobarbital anesthesia, and every effort was made to minimize suffering.

### Decision letter and Author response

Decision letter https://doi.org/10.7554/eLife.58949.sa1
Author response https://doi.org/10.7554/eLife.58949.sa2

## Additional files

**Supplementary files**

• Source data 1. Demographics and serum metabolic parameter measurements in Human Studies 1-4.

• Supplementary file 1. Tables containing characteristics of participants of the human studies. (**A**) Characteristics of participants of the 1st and 2nd study. (**B**) Characteristics of study participants of the 3rd study. (**C**) Characteristics of participants of the 4th study.

• Transparent reporting form

**Data availability**

All data generated or analysed during this study are included in the manuscript and supporting files. Source data files have been provided for all figures and figure supplements.

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
