## [Decision Letter]

**Acceptance summary:**

You provide convincing evidence that LCN2 is a satiety factor, as well as an anorexigenic signal and hunger indicator in primates. Furthermore, the failure to stimulate postprandial LCN2 in individuals with obesity may contribute to metabolic dysregulation. The studies are generally applicable to understanding the biology of satiety and hunger across species. All issues have been thoroughly and thoughtfully addressed.

**Decision letter after peer review:**

Thank you for submitting your article "Lipocalin-2 is a hunger biomarker and anorexigenic signal in primates" for consideration by *eLife*. Your article has been reviewed by three peer reviewers, and the evaluation has been overseen by Mone Zaidi as the Reviewing Editor and Clifford Rosen as the Senior Editor. The reviewers have opted to remain anonymous.

The reviewers have discussed the reviews with one another and the Reviewing Editor has drafted this decision to help you prepare a revised submission.

Summary:

Previous work by the same group has established in mice that lipocalin-2 (LCN-2) is a bone-derived hormone suppressing food intake and acting as a satiety signal (Mosialou et al., 2015). Yet, whether LCN-2 fulfils the same function in primates, including humans, remains unknown. In the present manuscript, the authors address this important gap in knowledge through multiple approaches. They show that circulating LCN2 increases postprandially in non-obese subjects, and that the serum levels of LCN2 negatively correlates with hunger sensation after a meal challenge. They provide evidence for the binding of radiolabeled or biotinylated recombinant human LCN2 in the hypothalamus of monkeys and humans. They also find that daily treatment of monkeys with recombinant human LCN2 decreases food intake by about 20%. This is an important study; the paper is generally well-written, and the data are convincing. With that said, certain issues need to be addressed and the manuscript modified accordingly.

Essential revisions:

1) While this paper provides compelling evidence that LCN2 is a satiety signal, it is less clear that it is a biomarker of hunger. Please comment.

2) A weakness seems to be a lack of statistical power in some of the studies, in particular in the experiments involving the injection of LCN2 into monkey. Some results don't reach statistical significance because only 3 monkeys were use in each group (Figure 4B-D, F, and Figure 5B and D). While the reviewers recognize that it may not be feasible to increase monkey numbers due to a number of reasons, an explanation to account for the lack of power is required, and this caveat should be clearly stated.

3) In the study presented in Figure 4, monkeys are given a daily injection of LCN2 and the authors show that the circulating level of LCN2 is increased 4 h after the injection (Figure 4F). Have they formally determined the half-life of recombinant human LCN2? In other words, do they have data (or samples on which they can measure circulating levels of LCN2 at earlier time points, namely 10, 15, 30, 60 or 120 minutes after the injection? If not, it is suggested that they conduct a formal pharmacokinetic study and post the results onto BioRxiv. Furthermore, and of note, the authors argue that the postprandial increase in LCN2 observed in the average-weight humans (16-50%) is in the same order of magnitude than the one they previously observed in mice (Mosialou et al., 2017). However, in their previous paper the postprandial increase in mice was about 300% within the first 3hrs. This difference should be discussed appropriately in the Discussion.

4) Please clarify whether the monkeys were injected just before they received food or whether feeding was ad libitum?

5) The daily dose of recombinant LCN2 injected in the monkey in Figure 4 is not indicated in the Materials and methods or anywhere else in the manuscript. How was the dose determined? Have they performed preliminary studies to determine which doses produce a significant raise in circulating LCN2? Please therefore explain the strategy to determine the optimal dose.

6) In their previous studies in mice, the recombinant mouse LCN2 was produced in bacteria. Here they use a mammalian cell expression system, which is certainly a good idea to ensure the presence of all the posttranslational modifications normally present in LCN2 such as N-glycosylation (Kjeldsen et al. JBC 1993). It will be important to show the purity and the integrity of their recombinant protein on an SDS-PAGE gel (Coomassie) and to confirm that it is indeed N-glycosylated. In addition, have they verified the absence of endotoxin in their preparation of recombinant human LCN2? These are simple biochemical analyses that could be accomplished with ease and would strengthen the developmentability of their molecule.

7) Autoradiography studies are underpowered (e.g., 1 brain slice and 4 fields of view) for quantitative assessment of %LCN positive cells (Figure 2V-W, Figure 4—figure supplement 1D-F). The reviewers suggest either using more biological replicates or stating that these are qualitative representations.

---

## [Author Response]

Essential revisions:1) While this paper provides compelling evidence that LCN2 is a satiety signal, it is less clear that it is a biomarker of hunger. Please comment.

This is a valid point. In the absence of additional data to further support the notion that LCN2 is a biomarker of hunger we have removed it from the title and throughout the text. We now clarify that LCN2 is a satiety signal.

2) A weakness seems to be a lack of statistical power in some of the studies, in particular in the experiments involving the injection of LCN2 into monkey. Some results don't reach statistical significance because only 3 monkeys were use in each group (Figure 4B-D, F, and Figure 5B and D). While the reviewers recognize that it may not be feasible to increase monkey numbers due to a number of reasons, an explanation to account for the lack of power is required, and this caveat should be clearly stated.

It should be noted that in Figure 5B and D (previous 4B and D) the results show a trend towards reduction of food intake in the LCN2-treated animals. However, they do not reach statistical significance because of the small number of animals per group (N=3). In Figure 5E (previous 4E) which combines the data from the two treatment weeks (N=6 animals/group) the observed reduction in food intake is statistically significant.

Two different ELISA assays were used; one for human and one for monkey LCN2. Each ELISA has selective reactivity for the designated species. Therefore, in Figure 5F (previous 4F) gray bars indicate serum levels of monkey LCN2 whereas red bars represent serum levels of human LCN2 following its administration to N=6 monkeys. Even though the combined human and monkey LCN2 levels approximately doubled following administration of recombinant human LCN2, this increase was not statistically significant (p=0.35). Similarly, Figure 6D and E (previous 5B and D) show a trend towards decrease in body weight and serum Leptin levels but those were not statistically significant. We believe that the lack of power in these data reflects the limited number of monkeys used in each group (N=3). These caveats are now clearly stated in the Results text describing these data.

3) In the study presented in Figure 4, monkeys are given a daily injection of LCN2 and the authors show that the circulating level of LCN2 is increased 4 h after the injection (Figure 4F). Have they formally determined the half-life of recombinant human LCN2? In other words, do they have data (or samples on which they can measure circulating levels of LCN2 at earlier time points, namely 10, 15, 30, 60 or 120 minutes after the injection? If not, it is suggested that they conduct a formal pharmacokinetic study and post the results onto BioRxiv.

We are in the process of completing a pharmacokinetic study in monkeys to determine the half-life and kinetics of administered LCN2. We will post these results at bioRxiv as per the reviewer’s suggestions.

Furthermore, and of note, the authors argue that the postprandial increase in LCN2 observed in the average-weight humans (16-50%) is in the same order of magnitude than the one they previously observed in mice (Mosialou et al., 2017). However, in their previous paper the postprandial increase in mice was about 300% within the first 3hrs. This difference should be discussed appropriately in the Discussion.

We now clearly indicate in the Discussion that whereas the postprandial increase in LCN2 observed in the average-weight humans was 16-50% the one we previously observed in mice was approximately 300% within the first 3hrs (Mosialou et al., 2017). This difference could be due to several reasons: (1) Difference in meal patterns between the two species. Mice mainly eat in the night and overnight fasting would result in approximately 75% food deprivation. Humans fast after midnight, which would be after dinner, and were offered a meal at or around breakfast time. Thus, fasting and hunger in humans would not be as severe as in mice and could affect the magnitude of postprandial LCN2 response after feeding. (2) Differences in the amount of food consumed within the same time after fast/refeed between mice and humans. Mice consume approximately 40% (1.6g) of their daily food intake (4g total on average) within 3h. In our study 1, 2 and 3, humans consumed 1.37% (732kcal), 10.6% (212kcal) and 14% (275kcal) of their daily food intake (2000 calories/day, https://health.gov/our-work/food-nutrition/20152020-dietary-guidelines/guidelines/appendix-2/), respectively. These numbers account for a 2.8- to 29–fold higher food intake in mice than in humans within 3 hours. Higher food intake in mice may have stimulated the higher increase in postprandial LCN2 levels.

4) Please clarify whether the monkeys were injected just before they received food or whether feeding was ad libitum?

Animals were injected with rh-LCN2 or saline in the morning and then pairs were separated and fed individually for 4 hours. After the 4 hour feeding periods all food was removed and weighed and then pairs were reunited. This means that animals were essentially fasted each night. Some enrichment foods (small pieces of fruit or vegetables) were provided in the afternoon, but those accounted to be a low percentage of their daily caloric intake. On blood sampling days the animals had samples collected in the afternoon, after the 4-hour feeding period. This information is now provided in the Materials and methods under the section “Non-human primate study”.

5) The daily dose of recombinant LCN2 injected in the monkey in Figure 4 is not indicated in the Materials and methods or anywhere else in the manuscript. How was the dose determined? Have they performed preliminary studies to determine which doses produce a significant raise in circulating LCN2? Please therefore explain the strategy to determine the optimal dose.

The LCN2 dose administered in monkeys to evaluate effects on food intake was extrapolated from our studies in mice (Mosialou et al.*, 2*017). In the mouse hypothalamus the amount of naturally occurring LCN2 is 28 pg/mg and in the adult mouse and human serum it is on average 100-150 ng/ml. In mice, administration of LCN2 by intraperitoneal injection of 150 ng/g daily crosses the blood brain barrier and suppresses appetite. Using interspecies conversion per m^2^, we have calculated the monkey dose to be 0.0375 mg/kg. This dose is equivalent to the amount used to treat mice and it is calculated based on body surface area; it takes into account interspecies variation in several physiological parameters including oxygen utilization, caloric expenditure, basal metabolism and blood volume (Reagan-Shaw, Nihal and Ahmad, 2008). This information in now provided in Results under the section “rh-LCN2 treatment suppresses food intake and body weight in vervets within five days of treatment”.

6) In their previous studies in mice, the recombinant mouse LCN2 was produced in bacteria. Here they use a mammalian cell expression system, which is certainly a good idea to ensure the presence of all the posttranslational modifications normally present in LCN2 such as N-glycosylation (Kjeldsen et al. JBC 1993). It will be important to show the purity and the integrity of their recombinant protein on an SDS-PAGE gel (Coomassie) and to confirm that it is indeed N-glycosylated. In addition, have they verified the absence of endotoxin in their preparation of recombinant human LCN2? These are simple biochemical analyses that could be accomplished with ease and would strengthen the developmentability of their molecule.

We performed Coomassie blue staining in the same stocks of recombinant human LCN2 that were used for the experiments in non-human primates. Stocks were digested with N-glycanase to examine N-glycosylation. Figure 5—figure supplement 1 shows that the recombinant protein is pure and N-glycosylated. In addition, in the same stocks we measured endotoxin levels by a chromogenic LAL assay and found them to be 1EU/mg of protein. We then used an endotoxin removal kit which lowered the endotoxin levels by 20% (0.8EU/mg of protein). According to the U.S. Food and Drug Administration (https://www.fda.gov/inspections-compliance-enforcementand-criminal-investigations/inspection-technical-guides/bacterial-endotoxinspyrogens) the endotoxin limit for the parenteral (in our case intravenous injection) administration of drugs is 5 EU/kg/hr. Based on measured endotoxin content of our preparation and the body weights of the non-human primates at the time of LCN2 administration, each animal received a maximum endotoxin of 0.03 EU/kg/hr. This information is provided in the Materials and methods under the section “Production of recombinant human LCN2 (rh-LCN2)” and in Results under the section “rh-LCN2 treatment suppresses food intake and body weight in vervets within five days of treatment”.

7) Autoradiography studies are underpowered (e.g., 1 brain slice and 4 fields of view) for quantitative assessment of %LCN positive cells (Figure 2V-W, Figure 4—figure supplement 1D-F). The reviewers suggest either using more biological replicates or stating that these are qualitative representations.

Bar graphs were obtained from a single brain section and therefore depict qualitative representations of binding. This clarification is provided in the text and legend of Figure 4 and Figure 4—figure supplement 1.